# Learning-Augmented Streaming Algorithms for Correlation Clustering

**Yinhao Dong[1]    Shan Jiang[1]    Shi Li[2,3]    Pan Peng[1]**[*]

[1]School of Computer Science and Technology,
University of Science and Technology of China, Hefei, Anhui Province, China
[2]School of Computer Science, Nanjing University, Nanjing, Jiangsu Province, China
[3]New Cornerstone Science Laboratory

`{yhdong,js_}@mail.ustc.edu.cn, shili@nju.edu.cn, ppeng@ustc.edu.cn`

## Abstract

We study streaming algorithms for Correlation Clustering. Given a graph as an arbitrary-order stream of edges, with each edge labeled as positive or negative, the goal is to partition the vertices into disjoint clusters, such that the number of disagreements is minimized. In this paper, we give the first learning-augmented streaming algorithms for the problem on both complete and general graphs, improving the best-known space-approximation tradeoffs. Based on the works of Cambus et al. (SODA'24) and Ahn et al. (ICML'15), our algorithms use the predictions of pairwise distances between vertices provided by a predictor. For complete graphs, our algorithm achieves a better-than-3 approximation under good prediction quality, while using $\tilde{O}(n)$ total space. For general graphs, our algorithm achieves an $O(\log |E^-|)$ approximation under good prediction quality using $\tilde{O}(n)$ total space, improving the best-known non-learning algorithm in terms of space efficiency. Experimental results on synthetic and real-world datasets demonstrate the superiority of our proposed algorithms over their non-learning counterparts.

## 1 Introduction

Correlation Clustering is a fundamental problem in machine learning and data mining, and it has a wide range of applications, such as image segmentation [66], community detection [86, 91], automated labeling [28], etc. Given a graph $G = (V, E = E^+ \cup E^-)$, where each edge is labeled as positive $(+)$ or negative $(-)$, the goal is to find a clustering $\mathcal{C}$, i.e., a partition of $V$ into disjoint clusters $C_1, C_2, \ldots, C_t$, where $t$ is arbitrary, that minimizes the following cost:

$$\text{cost}_G(\mathcal{C}) := |\{(u,v) \in E^+ : \exists i \neq j, \ u \in C_i, v \in C_j\}| + |\{(u,v) \in E^- : \exists i, \ u,v \in C_i\}|. \quad (1)$$

That is, the number of positive edges between different clusters, plus the number of negative edges within the same cluster. (We often refer to this as the number of *disagreements*.)

The most commonly studied version of this problem is on complete graphs, introduced by Bansal et al. [15], and known to be NP-hard and even APX-hard [31]. Hence, significant efforts have been dedicated to designing approximation algorithms for this setting [5, 15, 23, 24, 31, 32, 37–39], culminating in a $1.437$-approximation via a linear program (LP) based rounding [23, 24]. In contrast, the problem on general graphs has received less attention. Charikar et al. [31] and Demaine et al. [44] proposed an $O(\log n)$-approximation algorithm via ball-growing based LP rounding. They also showed that this version is equivalent to the minimum multicut problem, and is thus APX-hard and unlikely to admit better-than-$\Theta(\log n)$ approximations.

---

[*]Corresponding Author.

39th Conference on Neural Information Processing Systems (NeurIPS 2025).

Partially due to storage limitations and the rapidly growing volume of data, *streaming algorithms* for Correlation Clustering have received increasing attention recently. In this setting, a graph is represented as a sequence of edge insertions or deletions, known as a *graph stream*. The objective is to scan the sequence in a few number of passes, ideally, 1 pass and find a high-quality clustering of the vertex set, while minimizing space usage. If the sequence contains only edge insertions, it is referred to as an *insertion-only* stream; if both insertions and deletions are allowed, it is referred to as a *dynamic* stream. Since the output of the clustering inherently requires $\Omega(n)$ space (as each vertex needs a label to indicate its cluster membership), most previous research has primarily focused on the *semi-streaming* model [50], i.e., the algorithm is allowed to use $\tilde{O}(n) := O(n \operatorname{polylog} n)$ space.[1] We note that the space used by the streaming algorithm can be further divided into the space for updating data structures during the stream and the space for post-processing.

A long line of prior work has focused on the complete graph setting [4, 10, 12, 17, 18, 22, 29, 36, 39]. If the total space of the algorithm is restricted to $\tilde{O}(n)$, then the best-known space-approximation tradeoff in dynamic streams is a $(3 + \varepsilon)$-approximation algorithm [22], which uses $\tilde{O}(\varepsilon^{-1} n)$ total space. If only the space used for updating during the stream is restricted to $\tilde{O}(n)$, then the best-known space-approximation tradeoff in dynamic streams is achieved by an $(\alpha_{\mathrm{BEST}} + \varepsilon)$-approximation algorithm [10], where $1.042 \leq \alpha_{\mathrm{BEST}} \leq 1.437$ [24] denotes the best approximation ratio of any polynomial-time classical algorithm. However, this algorithm does not bound the space for post-processing, which can be significantly larger. For general graphs, the only known dynamic streaming algorithm is due to Ahn et al. [4], which uses the multiplicative weight update method on the sparsified graph to achieve a $3(1 + \varepsilon) \log |E^-|$-approximation while using $\tilde{O}(\varepsilon^{-2} n + |E^-|)$ space.

In this paper, we explore new approaches that enable improved space-approximation tradeoffs on both complete and general graphs by leveraging ideas from *learning-augmented algorithms*. A learning-augmented algorithm uses *predictions* to enhance its performance. These algorithms stem from practical scenarios where machine learning techniques exploit data structure to exceed the worst-case guarantees of traditional algorithms. Our learning-augmented algorithms fit into the category of learning-augmented streaming algorithms [1, 2, 33, 47, 58, 61]. It is worth mentioning that both our work and previous efforts on learning-augmented streaming algorithms mainly focus on using predictors to improve the corresponding space-accuracy tradeoffs.

Now, we describe the prediction we are considering. We assume that the algorithm has oracle access to a predictor $\Pi : \binom{V}{2} \to [0, 1]$ that predicts the *pairwise distances* [2] $d_{uv}$ between any two vertices $u$ and $v$ in $V$. We believe such predictors are natural and arise in many situations.

**Example 1.1** (Multiple graphs on the same vertex set). It is common to define *multiple* graphs over the same vertex set. For example, in healthcare, patients can be represented as vertices, and different networks may capture relationships such as shared medical conditions (disease networks), visits to the same providers (provider networks), or participation in clinical trials. In biology, vertices might represent genes or proteins, with different networks reflecting protein-protein interactions, gene co-expression, or signaling pathways. Machine learning or data mining techniques can then be used to learn pairwise distances between nodes across these networks. If two patients or genes/proteins are similar in one network, they often exhibit similar behavior in others.

**Example 1.2** (Temporal graphs). A similar situation occurs in temporal graphs, where a sequence of graphs shares the same vertex set but has different edge sets over time. Pairwise distances learned from past graphs can help extract structural insights in the present or future.

*Leveraging these distances across networks can greatly aid in exploring the cluster structure of any newly defined network over the same vertex set.* We also remark that several other works have considered similar oracles for pairwise distance in different contexts, such as the query model [67, 87].

## 1.1 Our results

Our results are summarized in Table 1. Using the above predictions, we give the first learning-augmented streaming algorithms for Correlation Clustering on both complete and general graphs. Specifically, for complete graphs, our algorithm beats the 3-approximation if the predictions are

---

[1] On the other hand, Assadi et al. [11] studied streaming algorithms using $\operatorname{polylog} n$ bits of space for estimating the optimum Correlation Clustering *cost*, while their algorithms do *not* find the clustering.

[2] Note that one can directly treat $1 - d_{uv}$ as the pairwise *similarity* between $u$ and $v$.

Table 1: Comparison of our main results with the best-known space-approximation tradeoffs in polynomial time. Here, $n = |V|, \varepsilon \in (0,1)$ and $\beta \geq 1$. All algorithms are single-pass, and space is measured in words.

| Setting | Best-known space-approximation tradeoffs (without predictions) | Our results |
|---|---|---|
| complete graphs, dynamic streams | $(3 + \varepsilon)$-approximation, $\tilde{O}(\varepsilon^{-1}n)$ total space [22] | $(\min\{2.06\beta, 3\} + \varepsilon)$-approximation, $\tilde{O}(\varepsilon^{-2}n)$ total space (Theorem 1.3) |
| | $(\alpha_{\text{BEST}} + \varepsilon)$-approximation, $\tilde{O}(\varepsilon^{-2}n)$ space for updating, $\text{poly}(n)$ space for post-processing [10] | |
| general graphs, dynamic streams | $O(\log |E^-|)$-approximation, $\tilde{O}(\varepsilon^{-2}n + |E^-|)$ total space [4] | $O(\beta \log |E^-|)$-approximation, $\tilde{O}(\varepsilon^{-2}n)$ total space (Theorem 1.4) |

good, while still achieving a $(3 + \varepsilon)$-approximation even if the predictor behaves poorly. For general graphs, our algorithm achieves an $O(\log |E^-|)$-approximation under good quality while using less space than the existing non-learning algorithm. Furthermore, our algorithms are simple and easily implementable. We will use a parameter $\beta \geq 1$ to measure the quality of our predictor. Informally, we call a predictor $\beta$-*level* if the cost of the clustering induced by the predictions is at most a $\beta$ factor of the cost of the optimal solution. We refer to Section 3 for the formal definition. In short, the smaller $\beta$ is, the higher the quality of the predictor.

For the complete graph setting, we have the following theorem. (In the following and throughout the paper, "with high probability" refers to the probability of at least $1 - 1/n^c$ for some constant $c > 0$.)

**Theorem 1.3.** *Let $\varepsilon \in (0, 1/4)$ and $\beta \geq 1$. Given oracle access to a $\beta$-level predictor, there exists a single-pass streaming algorithm that, with high probability, achieves an expected $(\min\{2.06\beta, 3\}+\varepsilon)$-approximation for Correlation Clustering on complete graphs in dynamic streams. The algorithm uses $\tilde{O}(\varepsilon^{-2}n)$ words of space.*

Note that our algorithm achieves a better-than-3 approximation in dynamic streams under good prediction quality, whereas the previous best-known semi-streaming algorithm in dynamic streams is a $(3 + \varepsilon)$-approximation due to Cambus et al. [22]. That is, our algorithm is both consistent and robust, as desired for most natural learning-augmented algorithms [80]. We further highlight that the recent $(\alpha_{\text{BEST}} + \varepsilon)$-approximation algorithm [10] does not bound the space usage during the post-processing phase, which may be larger than $\tilde{O}(n)$. On the other hand, while there exists a single-pass $(1 + \varepsilon)$-approximation algorithm for dynamic streams [4, 18], it requires exponential post-processing time and is thus impractical. In contrast, our algorithm uses polynomial post-processing time.

Furthermore, we also obtain an algorithm for complete graphs in insertion-only streams (see Appendix E), which differs from our algorithm in dynamic streams but achieves the same approximation ratio. It is simpler and more practical than the existing 1.847-approximation algorithm [39].

For general graphs, we present the following result on the approximation-space trade-off for streaming Correlation Clustering, using a slightly different type of predictor.

**Theorem 1.4.** *Let $\varepsilon \in (0, 1/4)$ and $\beta \geq 1$. Given oracle access to an adapted $\beta$-level predictor, there exists a single-pass streaming algorithm that, with high probability, achieves an $O(\beta \log |E^-|)$-approximation for Correlation Clustering on general graphs in dynamic streams. The algorithm uses $\tilde{O}(\varepsilon^{-2}n)$ words of space.*

We remark that the above algorithm can be extended to work under the previous notion of predictors for a broad class of graphs (see Corollary F.1). Note that the best-known streaming algorithm for general graphs is an $O(\log |E^-|)$-approximation while using $\tilde{O}(\varepsilon^{-2}n + |E^-|)$ words of space [4]. In contrast, our learning-augmented algorithm attains a comparable approximation ratio under good prediction quality, while using less space. We also note that it is standard to assume that the space required to implement/store the predictor is *not* included in the space usage of our algorithms, as is common in learning-augmented streaming algorithms [1, 2, 33, 47, 58, 61].

To complement our theoretical results, we conduct comprehensive experiments to evaluate our algorithms on both synthetic and real-world datasets. Experimental results demonstrate the superiority of our learning-augmented algorithms. All the missing proofs are deferred to appendix.

## 1.2 Further related work

**Correlation Clustering.** In this paper, we study streaming algorithms for Correlation Clustering. In the era of big data, other sublinear models for this problem have also received considerable attention in recent years, including sublinear-time algorithms [12, 23, 39], the Massively Parallel Computation (MPC) model [17, 23, 25, 36, 39, 41], and vertex/edge fully dynamic models [16, 41]. From another perspective, we focus on the minimization version of Correlation Clustering, which aims to minimize the number of disagreements. Other variants of this problem have also been studied, such as maximizing agreements [89], or minimizing certain norms—or all norms—of the disagreement vector [26, 30, 42, 56, 63, 82].

**Learning-augmented algorithms.** Learning-augmented algorithms, also known as algorithms with predictions, have been actively studied in the context of online algorithms [6, 7, 9, 13, 59, 68, 77, 83], data structures [51, 73, 79, 85, 90], graph algorithms [14, 20, 34, 43, 45, 46, 57, 69, 74], sublinear-time algorithms [48, 60, 72], approximation algorithms [8, 21, 35, 49, 53, 81], and mechanism design [3, 27, 40, 54, 76, 92], among others. Our work fits into the category of learning-augmented (graph) streaming algorithms.

## 2 Preliminaries

**Notations.** Throughout the paper, we let $G = (V, E)$ be an undirected and unweighted graph with $|V| = n$, $|E| = m$, where each edge is labeled as positive or negative (i.e., $E = E^+ \cup E^-$). In some places of the paper, we identify the input graph only with the set of positive edges, i.e., $G^+ = (V, E^+)$ and the negative edges are defined implicitly. For each vertex $u \in V$, let $N(u)$ be the set of all neighbors of $u$ and $N^+(u)$ be the set of positive neighbors of $u$ (i.e., vertices that are connected by a positive edge). Correspondingly, let $\deg(u) := |N(u)|$ be the degree of $u$, and similarly, $\deg^+(u) := |N^+(u)|$. We use $\text{cost}_G(\mathcal{C})$ to denote the cost of the clustering $\mathcal{C}$ on $G$, as defined in Eq. (1). We say an algorithm achieves an $\alpha$-approximation if it outputs a clustering $\mathcal{C}$ on $G$ such that $\text{OPT} \leq \text{cost}_G(\mathcal{C}) \leq \alpha \cdot \text{OPT}$, where OPT denotes the cost of an optimal solution on $G$.

Due to space limitations, we introduce additional technical preliminaries in Appendix A.

## 3 The $\beta$-level predictor

In this section, we give the formal definition of a $\beta$-level predictor.

**Definition 3.1** ($\beta$-level predictor). For any $\beta \geq 1$, we call a predictor $\beta$-*level*, if it predicts the pairwise distances $d_{uv} \in [0, 1]$ between any two vertices $u$ and $v$ in $G$ such that

(1) (triangle inequality) $d_{uv} + d_{vw} \geq d_{uw}$ for all $u, v, w \in V$,

(2) $\sum_{(u,v) \in E^+} d_{uv} + \sum_{(u,v) \in E^-} (1 - d_{uv}) \leq \beta \cdot \text{OPT}$.

Intuitively, a smaller $\beta$ indicates a higher-quality predictor, and in this case $d_{uv}$ can be used to determine how likely $u$ and $v$ are in the same cluster of the optimal solution. However, we point out that the predictions can be completely independent of the input graph. In the worst case, the predictions can be arbitrarily bad, which is allowed for learning-augmented algorithms since robustness is a desired goal. We remark that Definition 3.1 is inspired by the metric LP formulation of Correlation Clustering. In a sense, the $\beta$-level predictor corresponds to a feasible solution to the LP.

Furthermore, for a general graph, we say a pairwise distance predictor is an *adapted $\beta$-level predictor* if it satisfies the triangle inequality and $\sum_{(u,v) \in E_H^+} w'_{uv} d_{uv} + \sum_{(u,v) \in E^-} (1 - d_{uv}) \leq \beta \cdot \text{OPT}$, where $H^+ := (V, E_H^+, w')$ is an $\varepsilon$-spectral sparsifier of $G^+ = (V, E^+)$, which approximates all the cuts in $G^+$ within a $(1 \pm \varepsilon)$ factor.

**Practical consideration of the predictor.** As mentioned earlier, to cluster a graph, we may use ML models such as graph neural networks (GNNs) to learn pairwise distances from related networks defined on the same vertex set. These models can be trained to extract meaningful distances, for example, by learning node embeddings that map vertices to a Euclidean space. The distances between these embeddings naturally serve as pairwise distances and satisfy the triangle inequality. Although the second condition in Definition 3.1 is a technical requirement for theoretical analysis,

our algorithms remain applicable even when the given pairwise distances do not strictly satisfy this condition. In practice, as long as the distances are meaningful, they can be directly incorporated into our framework. We refer to Section 6 for empirical results.

## 4 Our algorithm for complete graphs in dynamic streams

### 4.1 Offline version

**Overview.** To better illustrate the algorithmic ideas, we first describe the offline version of our algorithm (see Algorithm 3 in Appendix C). The overall framework is similar to the algorithm proposed by Cambus et al. [22]. Our algorithm takes $G^+ = (V, E^+)$ as input. Initially, we pick a random permutation $\pi$ over the vertices. Then we divide all vertices into interesting and uninteresting vertices based on the relationship between the rank and the positive degree of a vertex. Specifically, a vertex $u$ is uninteresting if $\pi_u \geq \tau_u$ where $\tau_u := \frac{c}{\varepsilon} \cdot \frac{n \log n}{\deg^+(u)}$ (or equivalently $\deg^+(u) \geq \sigma_u$ where $\sigma_u := \frac{c}{\varepsilon} \cdot \frac{n \log n}{\pi_u}$), and interesting otherwise. Here, $\varepsilon \in (0, 1/4)$ and $c$ is a universal large constant. We run two pivot-based algorithms on the subgraph $G_{\text{store}}$ induced by the set of interesting vertices and obtain two clusterings. Finally, we output the clustering with the lower cost.

Specifically, the two pivot-based algorithms used in the clustering phase are described as follows.

**Algorithm TRUNCATEDPIVOT.** This algorithm simulates the Parallel Truncated-Pivot algorithm by Cambus et al. [22] and produces the same clustering. This algorithm proceeds in iterations. Let $U^{(t)}$ denote the set of unclustered vertices in $G_{\text{store}}$ at the beginning of iteration $t$. Initially, all the interesting vertices are unclustered. At the beginning of iteration $t$, if $U^{(t)} \neq \emptyset$, then we pick the vertex $u$ from $U^{(t)}$ with the smallest rank. Then we mark it as a pivot and create a pivot cluster $S^{(t)}$ containing $u$ and all of its unclustered positive neighbors in $G_{\text{store}}$. At the end of iteration $t$, we remove all vertices clustered in this iteration from $U^{(t)}$. Then the algorithm proceeds to the next iteration. If $U^{(t)} = \emptyset$ at the beginning of iteration $t$, then we know that all the interesting vertices are clustered. Now it suffices to assign each uninteresting vertex to a cluster. Each uninteresting vertex $u$ joins the cluster of pivot $v$ with the smallest rank if $(u, v) \in E^+$ and $\pi_v < \tau_u$. Then each unclustered vertex $u \in V$ creates a singleton cluster. Finally, we output all pivot clusters and singleton clusters. We defer its pseudocode (Algorithm 4) to Appendix C.

**Algorithm TRUNCATEDPIVOTWITHPRED.** This algorithm has oracle access to a $\beta$-level predictor $\Pi$. The algorithm closely resembles Algorithm TRUNCATEDPIVOT. The differences are as follows: (1) At iteration $t$, we create a pivot cluster $S^{(t)}$ containing $u$ and add all the unclustered vertices $v$ in $G_{\text{store}}$ to $S^{(t)}$ with probability $(1 - p_{uv})$ independently, where $p_{uv} = f(d_{uv})$ and $d_{uv} = \Pi(u, v)$. If $(u, v) \in E^+$, then $f(d_{uv}) = f^+(d_{uv})$; otherwise $f(d_{uv}) = f^-(d_{uv})$. We set $f^+(x)$ to be 0 if $x < a$, $(\frac{x-a}{b-a})^2$ if $x \in [a, b]$, and 1 if $x > b$, where $a = 0.19$ and $b = 0.5095$; we set $f^-(x) = x$. (2) Each uninteresting vertex $u$ joins the cluster of pivot $v$ in the order of $\pi$ with probability $(1 - p_{uv})$ independently, if $\pi_v < \tau_u$. We defer its pseudocode (Algorithm 5) to Appendix C.

We have the following approximation guarantee of the offline algorithm.

**Lemma 4.1.** *Let $\varepsilon \in (0, 1/4)$ and $\beta \geq 1$. Given oracle access to a $\beta$-level predictor, the offline algorithm (Algorithm 3) achieves an expected $(\min\{2.06\beta, 3\} + \varepsilon)$-approximation.*

### 4.2 Implementation in dynamic streams

In this subsection, we implement the offline algorithm in dynamic streams, as shown in Algorithm 1. A key observation is that it suffices to store the positive edges incident to interesting vertices since we apply pivot-based algorithms on the subgraph induced by interesting vertices and then try to assign uninteresting vertices to pivot clusters. To this end, we maintain a certain number of $\ell_0$-samplers for each vertex, which can be achieved in dynamic streams [62]. As we will see in the analysis, the $\ell_0$-samplers allow us to recover the edges incident to all the interesting vertices with high probability. Thus we can simulate the clustering phase of the offline algorithm. Specifically, we simulate TRUNCATEDPIVOT and TRUNCATEDPIVOTWITHPRED using the stored information, and finally output the clustering with the lower cost.

Note that in the final step, the cost of a clustering cannot be exactly calculated, as our streaming algorithm cannot store the entire graph. To overcome this challenge, we borrow the idea from [18] and

---

**Algorithm 1** An algorithm for complete graphs in dynamic streams

---

**Input:** Graph $G^+ = (V, E^+)$ as an arbitrary-order dynamic stream of edges, oracle access to a $\beta$-level predictor $\Pi$

**Output:** Clustering/Partition of $V$ into disjoint sets

    ▷ **Pre-processing phase**

1: Pick a random permutation of vertices $\pi : V \to \{1, \dots, n\}$.
2: **for** each vertex $u \in V$ **do**
3:     Let $\deg^+(u) \leftarrow 0$. Mark $u$ as unclustered and interesting.
4:     Let $\sigma_u := \frac{c}{\varepsilon} \cdot \frac{n \log n}{\pi_u}$, where $c$ is a universal large constant.
5:     Initialize $10c \log n \cdot \sigma_u$ independent $\ell_0$-samplers (with failure probability $1/10$) for the adjacency vector of $u$ (the row of the adjacency matrix of $G^+$ that corresponds to $u$).

    ▷ **Streaming phase**

6: **for** each item $(e_i = (u, v), \Delta_i \in \{-1, 1\})$ in the dynamic stream **do**
7:     Update $\deg^+(u)$, $\deg^+(v)$ and all the $\ell_0$-samplers associated with $u$ and $v$.
8: Maintain an $\varepsilon$-spectral sparsifier $H^+$ for $G^+$ using the algorithm of Theorem A.6.

    ▷ **Post-processing phase**

9: A vertex $u$ marks itself uninteresting if $\deg^+(u) \geq \sigma_u$.
10: Retrieve all incident edges of interesting vertices (with high probability) using the $\ell_0$ samplers.
11: Let $G_{\text{store}}$ be the graph induced by the interesting vertices.
12: $\mathcal{C}_1 \leftarrow \text{TRUNCATEDPIVOT}(G^+, G_{\text{store}}, \pi)$
13: $\mathcal{C}_2 \leftarrow \text{TRUNCATEDPIVOTWITHPRED}(G^+, G_{\text{store}}, \pi, \Pi)$
14: $\widetilde{\text{cost}}_G(\mathcal{C}_1) \leftarrow \sum_{C \in \mathcal{C}_1} (\frac{1}{2} \partial_{H^+}(C) + \binom{|C|}{2} - \frac{1}{2} \sum_{u \in C} \deg^+(u))$
15: $\widetilde{\text{cost}}_G(\mathcal{C}_2) \leftarrow \sum_{C \in \mathcal{C}_2} (\frac{1}{2} \partial_{H^+}(C) + \binom{|C|}{2} - \frac{1}{2} \sum_{u \in C} \deg^+(u))$
16: $i \leftarrow \arg\min_{i=1,2} \{\widetilde{\text{cost}}_G(\mathcal{C}_i)\}$.
17: **return** $\mathcal{C}_i$

---

utilize the graph sparsification technique [64] to estimate the clustering cost. Specifically, during the dynamic stream, we maintain an $\varepsilon$-spectral sparsifier $H^+$ for $G^+$ using the algorithm of Theorem A.6. We also maintain the positive degree $\deg^+(u)$ for each vertex $u$. Then we can approximate the cost of a clustering up to a $(1 \pm \varepsilon)$-multiplicative error with high probability. The formal proof of Theorem 1.3 is deferred to Appendix D.1.

## 4.3 Analysis of the offline algorithm

Since the final clustering returned by the offline algorithm is the one with the lower cost between those produced by the two pivot-based algorithms, we start by analyzing the costs of these two clusterings. For ease of analysis, we separately examine the approximation ratios of the equivalent versions (Algorithms CKLPU-PIVOT and PAIRWISEDISS) that produce these two clusterings.

**Algorithm CKLPU-PIVOT** (Algorithm 4 in [22])**.** This algorithm proceeds in iterations. Let $U^{(t)}$ denote the set of unclustered vertices at the beginning of iteration $t$. Initially, we pick a random permutation $\pi$ over vertices, and all the vertices are unclustered. At the beginning of iteration $t$, let $\ell_t = \frac{c}{\varepsilon} \cdot \frac{n \log n}{t}$. Each unclustered vertex $v$ with $\deg^+(v) \geq \ell_t$ creates a *singleton cluster*. We pick the $t$-th vertex $u$ in $\pi$. If $u$ is unclustered, then we mark it as a pivot and create a *pivot cluster* $S^{(t)}$ containing $u$ and all of its unclustered positive neighbors. At the end of iteration $t$, we remove all vertices clustered in this iteration from $U^{(t)}$ and proceed to the next iteration. Finally, we output all pivot clusters and singleton clusters. We defer its pseudocode (Algorithm 6) to Appendix C.

**Algorithm PAIRWISEDISS.** This algorithm has oracle access to a $\beta$-level predictor $\Pi$. The only difference from Algorithm CKLPU-PIVOT is that at iteration $t$, we create a *pivot cluster* $S^{(t)}$ containing $u$ and add all unclustered vertices $v$ to $S^{(t)}$ with probability $(1 - p_{uv})$ independently, where $p_{uv} = f(d_{uv})$ and $d_{uv} = \Pi(u, v)$. We defer its pseudocode (Algorithm 7) to Appendix C.

**The offline algorithm as a combination of CKLPU-PIVOT and PAIRWISEDISS** We first show that the offline algorithm can be equivalently viewed as a combination of Algorithms CKLPU-PIVOT and PAIRWISEDISS, assuming the same randomness is used.

**Lemma 4.2** (Lemma 8 in [22])**.** *If the offline algorithm (Algorithm 3) and* CKLPU-PIVOT *use the same permutation* $\pi$*, then* TRUNCATEDPIVOT *and* CKLPU-PIVOT *output the same clustering.*

**Lemma 4.3.** *If the offline algorithm (Algorithm 3) and* PAIRWISEDISS *use the same permutation* $\pi$ *and predictions* $\{d_{uv}\}_{u,v\in V}$*, then* TRUNCATEDPIVOTWITHPRED *and* PAIRWISEDISS *output the same clustering with the same probability.*

### 4.3.1 The approximation ratios of CKLPU-PIVOT and PAIRWISEDISS

Now it suffices to analyze Algorithms CKLPU-PIVOT and PAIRWISEDISS separately. We follow the analysis framework in [22]. Specifically, we analyze the costs of pivot clusters and singleton clusters separately. For the former, we can directly apply the analysis of original pivot-based algorithms [5, 32]. Note that here we only need to focus on a subset of vertices (i.e., $V \setminus V_{\text{sin}}$ where $V_{\text{sin}}$ is the set of singletons). For the latter, we divide all the positive edges incident to singleton clusters (denoted as $E_{\text{sin}}$) into good edges ($E_{\text{good}}$) and bad edges ($E_{\text{bad}}$). Specifically, we define a positive edge incident to a singleton cluster to be good if the other endpoint was included in a pivot cluster *before* the singleton was created. Otherwise, the edge is bad. In other words, bad edges are those that either connect two singletons or the other endpoint was included in a pivot cluster *after* the singleton was created. The analysis in [22] shows that both the costs of good and bad edges can be charged to the pivot clusters, allowing us to bound the overall clustering cost.

The following lemma states the approximation guarantee of CKLPU-PIVOT, and thus that of the clustering returned by TRUNCATEDPIVOT.

**Lemma 4.4** ([22])**.** *Let* $\varepsilon \in (0, 1/4)$*. Let* $\mathcal{C}_1$ *denote the clustering returned by* TRUNCATEDPIVOT*, then* $\mathbb{E}[\text{cost}_G(\mathcal{C}_1)] \leq (3 + 12\varepsilon) \cdot \text{OPT} + \frac{1+4\varepsilon}{n^{\alpha-2}}$*, where* $\alpha := c/2 - 1 \gg 2$*.*

Next, we focus on the analysis of Algorithm PAIRWISEDISS.

**Lemma 4.5.** *Let* $P$ *denote the cost of pivot clusters returned by Algorithm* PAIRWISEDISS*. We have* $\mathbb{E}[P] \leq 2.06\beta \cdot \text{OPT}$*.*

*Proof.* Consider iteration $t$ of PAIRWISEDISS, if vertex $u$ considered in this iteration is unclustered (i.e., $u \in U^{(t)}$), then we call iteration $t$ a *pivot iteration*. The key observation is that the pivot iterations in PAIRWISEDISS are equivalent to the iterations of 2.06-approximation LP rounding algorithm [32]: given that $u$ is unclustered (i.e., $u \in U^{(t)}$), the conditional distribution of $u$ is uniformly distributed in $U^{(t)}$, and the cluster created during this iteration contains $u$ and all the unclustered vertices $v$ added with probability $(1 - p_{uv})$. Therefore, we can directly apply the triangle-based analysis in [32]. Define $L := \sum_{(u,v)\in E^+} d_{uv} + \sum_{(u,v)\in E^-}(1 - d_{uv})$. Since the predictor is $\beta$-level, by Definition 3.1, the predictions $\{d_{uv}\}_{u,v\in V}$ satisfy triangle inequality and $L \leq \beta \cdot \text{OPT}$. It follows that for all pivot iterations $t$, $\mathbb{E}[P^{(t)}] \leq 2.06 \cdot \mathbb{E}[L^{(t)}]$, where $P^{(t)}$ is the cost induced by the pivot cluster created at iteration $t$, and $L^{(t)} := \sum_{(u,v)\in E^+\cap E^{(t)}} d_{uv} + \sum_{(u,v)\in E^-\cap E^{(t)}}(1 - d_{uv})$ where $E^{(t)}$ is the set of edges *decided* at iteration $t$, i.e., $E^{(t)} = \{(u,v) \in E : u, v \in U^{(t)}; u \in S^{(t)} \text{ or } v \in S^{(t)}\}$. By linearity of expectation, we have $\mathbb{E}[P] = \sum_{t \text{ is a pivot iteration}} \mathbb{E}[P_2^{(t)}] \leq 2.06 \cdot L \leq 2.06\beta \cdot \text{OPT}$. □

**Corollary 4.6.** *Let* $\varepsilon \in (0, 1/4)$*. Let* $\mathcal{C}_2$ *denote the clustering returned by* TRUNCATEDPIVOTWITH-PRED*. We have* $\mathbb{E}[\text{cost}_G(\mathcal{C}_2)] \leq (2.06\beta + 8.24\beta\varepsilon) \cdot \text{OPT} + \frac{1+4\varepsilon}{n^{\alpha-2}}$*, where* $\alpha := c/2 - 1 \gg 2$*.*

We defer the proofs of Lemma 4.3, Corollary 4.6 and, finally, Lemma 4.1 to Appendix D.

## 5 Our algorithm for general graphs in dynamic streams

**Overview of the algorithm** Our algorithm is given in Algorithm 2. The core of our algorithm builds upon the ball-growing framework in the work of Charikara et al. [31] and Demaine et al. [44]. In our algorithm, we apply this framework to a sparsified graph and use the predictions $d_{uv}$ as distance metrics. Specifically, during the streaming phase, we maintain an $\varepsilon$-spectral sparsifier $H^+ := (V, E_H^+, w')$ for $G^+ = (V, E^+)$.

At the same time, we store all arriving negative edges and track their space usage. If, at any point, this space exceeds $\tilde{O}(\varepsilon^{-2}n)$ words, we immediately stop storing negative edges. Once the stream ends (i.e., after $H^+$ has been constructed), we proceed to the post-processing phase and run the

**Algorithm 2** An algorithm for general graphs in dynamic streams

---

**Input:** Graph $G = (V, E)$ as an arbitrary-order dynamic stream of edges, oracle access to an adapted $\beta$-level predictor $\Pi$

**Output:** Clustering/Partition of $V$ into disjoint sets
    ▷ **Streaming phase**
1: Maintain an $\varepsilon$-spectral sparsifier $H^+ := (V, E_H^+, w')$ for $G^+$ using the algorithm of Theorem A.6.
2: Meanwhile, store all arriving negative edges and track their space usage. If the space ever exceeds $\tilde{O}(\varepsilon^{-2}n)$ words, then stop storing negative edges and, after the stream ends, **goto** Line 4.
    ▷ **Post-processing phase**
3: Run the post-processing phase of the algorithm by Ahn et al. [4] and **return** the clustering $\mathcal{C}_1$.
4: Let $V_R \leftarrow V$ and $E_R \leftarrow E_H^+$ denote the sets of remaining vertices and edges, respectively.
5: Let $\mathcal{C}_2 \leftarrow \emptyset$.
6: For any $u, v \in V$, $d_{uv} = \Pi(u, v)$.
7: **while** $V_R \neq \emptyset$ **do**
8:      Let $R := (V_R, E_R)$ denote the current graph. Pick an arbitrary vertex $u \in V_R$. Let $r_u \leftarrow 0$.
9:      Increase $r_u$ and grow a ball $B_d(u, r_u)$ on $R$ such that $\partial_{H^+}(B_d(u, r_u)) \leq 3\ln(n+1) \cdot \mathrm{vol}_{H^+}(B_d(u, r_u))$.
10:      $\mathcal{C}_2 \leftarrow \mathcal{C}_2 \cup B_d(u, r_u)$.
11:      Remove the vertices in $B_d(u, r_u)$ from $V_R$ and the incident edges from $E_R$.
12: **return** $\mathcal{C}_2$

---

ball-growing procedure on $H^+$ to obtain the final clustering $\mathcal{C}_2$. On the other hand, if the total space for storing negative edges remains within $\tilde{O}(\varepsilon^{-2}n)$ throughout the stream, we instead invoke the post-processing phase of the algorithm by Ahn et al. [4] and return the resulting clustering $\mathcal{C}_1$.

Next we describe the ball-growing procedure, which proceeds iteratively. We initialize the clustering $\mathcal{C}_2 = \emptyset$. Let $R := (V_R, E_R)$ denote the current graph at each iteration, where $V_R$ and $E_R$ are the sets of remaining vertices and edges, respectively. Initially, we set $V_R = V$ and $E_R = E_H^+$. At each iteration, we select an arbitrary vertex $u \in V_R$ as the center and initialize its radius as $r_u = 0$. We gradually increase $r_u$ to grow a ball around $u$ until a certain condition is met. Then we set all the vertices in the ball as a new cluster, remove them along with their incident edges from the current graph, and proceed to the next iteration. This process is repeated until no vertices are left.

More precisely, for a vertex $u \in V_R$ and a radius $r_u \geq 0$, we define the *ball* centered at $u$ with radius $r_u$ as $B_d(u, r_u) := \{v \in V_R : d_{uv} \leq r_u\}$, which consists of all vertices in $R^+$ within distance at most $r_u$ from $u$, where distances are measured according to the predictions.

We next define the cut value and volume associated with a ball. Let $\partial_{H^+}(B_d(u, r_u))$ denote the value of the cut $(B_d(u, r_u), V \setminus B_d(u, r_u))$ in $H^+$:

$$\partial_{H^+}(B_d(u, r_u)) := \sum_{\substack{(v,w) \in E_H^+: \\ v \in B_d(u,r_u), w \notin B_d(u,r_u)}} w'_{vw}.$$

Furthermore, we define the *volume* of the ball $B_d(u, r_u)$ in $H^+$, denoted by $\mathrm{vol}_{H^+}(B_d(u, r_u))$:

$$\mathrm{vol}_{H^+}(B_d(u, r_u)) := \frac{V^*}{n} + \sum_{\substack{(v,w) \in E_H^+: \\ v,w \in B_d(u,r_u)}} w'_{vw} d_{vw} + \sum_{\substack{(v,w) \in E_H^+: \\ v \in B_d(u,r_u), w \notin B_d(u,r_u)}} w'_{vw}(r_u - d_{uv}),$$

where $V^* := \sum_{(u,v) \in E_H^+} w'_{uv} d_{uv}$ denotes the total volume of the graph $H^+$.

The ball is finalized once the cut value induced by the ball is at most $O(\log n)$ times its volume. Specifically, we require that $\partial_{H^+}(B_d(u, r_u)) \leq 3\ln(n+1) \cdot \mathrm{vol}_{H^+}(B_d(u, r_u))$. This condition is guaranteed by the following lemma:

**Lemma 5.1** ([31, 44, 52]). *For any vertex $u$, there exists a radius $r_u < 1/3$ (which can be found in polynomial time) such that the corresponding ball $B_d(u, r_u)$ satisfies $\partial_{H^+}(B_d(u, r_u)) \leq 3\ln(n+1) \cdot \mathrm{vol}_{H^+}(B_d(u, r_u))$.*

## 5.1 Analysis of Algorithm 2

**Space complexity.** The space complexity of Algorithm 2 is dominated by maintaining the $\varepsilon$-spectral sparsifier and storing the negative edges during the streaming phase. By Theorem A.6 and the condition in Line 2, the algorithm uses $\tilde{O}(\varepsilon^{-2}n)$ words of space.

**Approximation guarantee.** Recall the condition in Line 2: if the space used to store negative edges remains within $\tilde{O}(\varepsilon^{-2}n)$ throughout the stream, then the resulting clustering $\mathcal{C}_1$ is given by the algorithm of Ahn et al. [4], which has the following guarantee.

**Lemma 5.2** ([4]). $\mathrm{cost}_G(\mathcal{C}_1) \le 3(1 + \varepsilon) \log |E^-| \cdot \mathrm{OPT} = O(\log |E^-|) \cdot \mathrm{OPT}$.

Otherwise, we have $|E^-| \ge n$, and the clustering $\mathcal{C}_2$ is obtained via the ball-growing procedure. We now analyze the cost of $\mathcal{C}_2$, which consists of two parts: the number of positive edges that cross between different clusters, and the number of negative edges that lie in the same cluster. In the following, we bound these two quantities separately.

Consider the weighted graph $H := (V, E_H^+ \cup E^-, \bar{w})$, where the edge weights are defined as $\bar{w}_{uv} = w'_{uv}$ for $(u, v) \in E_H^+$ and $\bar{w}_{uv} = 1$ for $(u, v) \in E^-$. The positive and negative costs of $\mathcal{C}_2$ on $H$, denoted $\mathrm{cost}_H^+(\mathcal{C}_2)$ and $\mathrm{cost}_H^-(\mathcal{C}_2)$ respectively, can be bounded by the following two lemmas.

**Lemma 5.3.** $\mathrm{cost}_H^+(\mathcal{C}_2) \le 3 \ln(n + 1) \cdot \sum_{(u,v) \in E_H^+} w'_{uv} d_{uv}$.

**Lemma 5.4.** $\mathrm{cost}_H^-(\mathcal{C}_2) \le 3 \sum_{(u,v) \in E^-} (1 - d_{uv})$.

Now we are ready to bound the cost of $\mathcal{C}_2$ on $G$.

**Lemma 5.5.** $\mathrm{cost}_G(\mathcal{C}_2) = O(\beta \log |E^-|) \cdot \mathrm{OPT}$.

Due to space limits, we defer the proofs of Lemma 5.3, Lemma 5.4 and Lemma 5.5 to Appendix F.

*Proof of Theorem 1.4.* Theorem 1.4 follows from Lemma 5.2 and Lemma 5.5. ☐

Furthermore, if the input graph satisfies certain mild conditions, then the adapted $\beta$-level predictor can be relaxed to the standard $\beta$-level predictor (Definition 3.1). We defer the details to Appendix F.4.

## 6 Experiments

In this section, we evaluate our proposed algorithm for complete graphs empirically on synthetic and real-world datasets. All experiments are conducted on a CPU with an i7-13700H processor and 32 GB RAM. For all results, unless otherwise stated, we report the average clustering cost over 20 independent trials. Our source code is available in the supplementary material.

**Datasets. 1) Synthetic datasets.** These datasets are generated from the Stochastic Block Model (SBM). We use this model to plant ground-truth clusters. It samples positive edges between vertex pairs within the same cluster with probability $p > 0.5$, and samples positive edges across different clusters with probability $(1 - p)$. **2) Real-world datasets.** We use EMAILCORE [70, 94], FACE-BOOK [78], LASTFM [84], and DBLP [93] datasets. For simplicity, for all datasets, we only simulate insertion-only streams of edges. We refer to Appendix G.1 for detailed descriptions of the datasets.

**Predictor descriptions. 1) Noisy predictor.** We use this predictor for datasets with available optimal clusterings. We form this predictor by performing perturbations on optimal clusterings. **2) Spectral embedding.** We use this predictor for EMAILCORE and LASTFM. It first maps all vertices to $\mathbb{R}^d$ using the graph Laplacian, then clusters all vertices based on their embeddings. For any two vertices $u, v \in V$, we form the prediction $d_{uv}$ based on the spectral embeddings of $u, v$. **3) Binary classifier.** We use this predictor for datasets with available ground-truth communities. This predictor is constructed by training a binary classifier to predict whether two vertices belong to the same cluster using node features. The predictions (i.e., binary values in $\{0, 1\}$) are then used as pairwise distances $d_{uv}$ in our algorithms. We refer to Appendix G.2 for detailed descriptions of the predictors.

**Baselines. 1)** $(3 + \varepsilon)$**-approximation non-learning counterparts.** For our algorithm for complete graphs in dynamic streams, the counterpart is Algorithm CKLPU24 [22]. **2) The agreement decomposition algorithm** CLMNPT21 [36]. Though the approximation ratio in theory is large

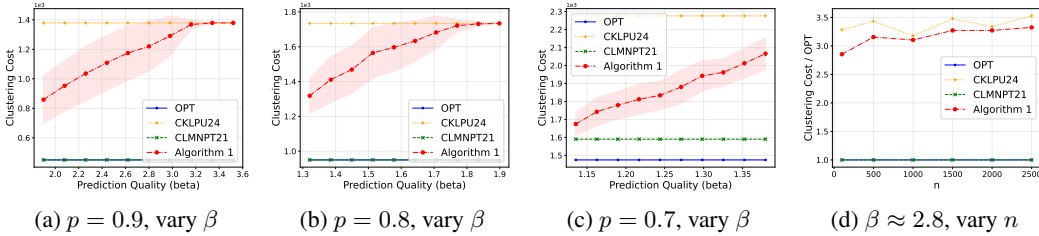

(a) $p = 0.9$, vary $\beta$  (b) $p = 0.8$, vary $\beta$  (c) $p = 0.7$, vary $\beta$  (d) $\beta \approx 2.8$, vary $n$

Figure 1: Performance of Algorithm 1 on synthetic datasets. We examine the effects of prediction quality $\beta$, SBM parameter $p$, and graph size $n$. We set $n = 100$ in (a)–(c) and $p = 0.95$ in (d).

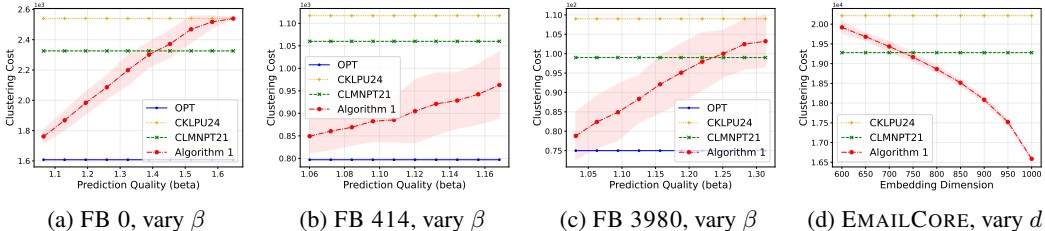

(a) FB 0, vary $\beta$  (b) FB 414, vary $\beta$  (c) FB 3980, vary $\beta$  (d) EMAILCORE, vary $d$

Figure 2: Performance of Algorithm 1 on real-world datasets. (a)–(c) show the effect of $\beta$ on three FACEBOOK subgraphs. (d) shows the effect of the dimension $d$ of spectral embeddings on EMAILCORE. Note that a larger $d$ indicates higher prediction quality (i.e., a smaller $\beta$).

Table 2: Clustering costs ($\times$ 1e3) of Algorithm 1 with binary classifiers as predictors, compared to its non-learning counterpart. The reported values are averaged over 5 runs.

| Algorithm ＼ Dataset | SBM ($n = 1200$) | SBM ($n = 2400$) | SBM ($n = 3600$) | DBLP |
|---|---|---|---|---|
| CKLPU24 | 105.3 | 524.8 | 1 114.3 | 537.5 |
| **Algorithm 1** | 35.9 | 145.6 | 324.9 | 247.1 |

($\approx 701$), this algorithm has been shown to give high-quality solutions in practice. Note that this algorithm requires multiple passes. For fairness, all baselines were implemented with equal effort.

**Results on synthetic datasets.** Figure 1 shows the performance of Algorithm 1 on synthetic datasets. **1) Varying $\beta$ and $p$.** We first examine the effect of $\beta$ and $p$ (see Figures 1(a)–(c)). When $\beta$ is small, the cost of our algorithm is significantly lower than that of the $(3 + \varepsilon)$-approximation non-learning baseline. Even for large $\beta$, our algorithm performs no worse. Notably, we observe that the algorithm of CLMNPT21 outputs (near-)optimal solution. We attribute this to the fact that the SBM graphs contain many dense components, making them well-suited for this algorithm. Moreover, even when the ground-truth communities become less obvious (e.g., $p = 0.7$), the clustering cost of Algorithm 1 is reduced by up to 26% compared to the algorithm of CKLPU24. **2) Varying $n$.** Furthermore, we investigate whether our algorithm scales well with graph size (see Figure 1(d)). To clearly present our results, we calculate the ratio between the cost of each algorithm and the optimal solution. The result demonstrates that our algorithm performs well consistently as the graph size increases.

**Results on real-world datasets.** Figure 2 shows the performance of Algorithm 1 on real-world datasets. The results demonstrate that under good prediction quality, Algorithm 1 consistently outperforms other baselines across all datasets used. For example, in Figure 2(a), when $\beta \approx 1.2$, the average cost of our algorithm is 15% lower than that of CLMNPT21 and 22% lower than that of CKLPU24. Besides, in Figure 2(d), our algorithm reduces the clustering cost by up to 14% compared to CLMNPT21. Even in case of poor predictions, Algorithm 1 does not perform worse than its $(3 + \varepsilon)$-approximation counterpart without predictions.

**Results based on binary classifiers as predictors.** Table 2 shows the performance of Algorithm 1 with binary classifiers as predictors, a more realistic setting. These experiments are performed on three SBM graphs with parameter $p = 0.95$ and varying sizes, as well as the DBLP dataset (a sampled subgraph with 10 000 vertices). The results show that our learning-augmented algorithm consistently outperforms its non-learning counterpart across all datasets. For instance, on the SBM graph with 2 400 vertices, Algorithm 1 reduces the clustering cost by 72% compared to CKLPU24.

## Acknowledgments and Disclosure of Funding

The work of YD, SJ, and PP is supported in part by NSFC Grant 62272431 and the Innovation Program for Quantum Science and Technology (Grant No. 2021ZD0302901). The work of SL is supported by the State Key Laboratory for Novel Software Technology and the New Cornerstone Science Foundation.

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

# A Additional technical preliminaries

In this paper, we frequently use graph sparsification techniques, so we review the relevant definitions in this section.

$\ell_0$**-samplers.** We first review the definition of $\ell_0$-samplers.

**Definition A.1** ($\ell_0$-sampler [62]). Let $\boldsymbol{x} \in \mathbb{R}^n$ be a non-zero vector and $\delta \in (0, 1)$. An $\ell_0$-sampler for $\boldsymbol{x}$ returns FAIL with probability at most $\delta$ and otherwise returns some index $i$ such that $x_i \neq 0$ with probability $\frac{1}{|\mathrm{supp}(\boldsymbol{x})|}$ where $\mathrm{supp}(\boldsymbol{x}) = \{i \mid x_i \neq 0\}$ is the support of $\boldsymbol{x}$.

The following theorem states that $\ell_0$-samplers can be maintained using a single pass in dynamic streams.

**Theorem A.2** ([62]). *There exists a single-pass streaming algorithm for maintaining an $\ell_0$-sampler for a non-zero vector $\boldsymbol{x} \in \mathbb{R}^n$ (with failure pribability $\delta$) in the dynamic model using $O(\log^2 n \log \delta^{-1})$ bits of space.*

**Cut sparsifiers.** Cut sparsifiers [19] are a basic notion in graph sparsification.

**Definition A.3** (Cut sparsifier [19]). Let $H = (V_H, E_H, w)$ be an undirected graph and let $\varepsilon \in (0, 1)$. We say that a reweighted subgraph $H' = (V_H, E'_H, w')$ is an *$\varepsilon$-cut sparsifier* of $H$ if for any $A \subseteq V_H$,

$$(1 - \varepsilon)\partial_H(A) \leq \partial_{H'}(A) \leq (1 + \varepsilon)\partial_H(A),$$

where $\partial_H(A) := \sum_{(u,v) \in E_H : u \in A, v \notin A} w_{uv}$ and $\partial_{H'}(A) := \sum_{(u,v) \in E'_H : u \in A, v \notin A} w'_{uv}$ denote the weights of the cut $(A, V_H \setminus A)$ in $H$ and $H'$, respectively.

**Spectral sparsifiers.** Spectral sparsifiers [88] are a stronger notion than cut sparsifers.

**Definition A.4** (Spectral sparsifier [88]). Let $H = (V_H, E_H, w)$ be an undirected graph and let $\varepsilon \in (0, 1)$. We say that a reweighted subgraph $H' = (V_H, E'_H, w')$ is an *$\varepsilon$-spectral sparsifier* of $H$ if for any $\boldsymbol{x} \in \mathbb{R}^n$,

$$(1 - \varepsilon)\boldsymbol{x}^\top L_H \boldsymbol{x} \leq \boldsymbol{x}^\top L_{H'} \boldsymbol{x} \leq (1 + \varepsilon)\boldsymbol{x}^\top L_H \boldsymbol{x},$$

which is equivalent to

$$(1 - \varepsilon)L_H \preceq L_{H'} \preceq (1 + \varepsilon)L_H,$$

where $L_H$ and $L_{H'}$ denote the Laplacian matrix of $H$ and $H'$, respectively.

It is easy to see that if $H'$ is an $\varepsilon$-spectral sparsifier of $H$, then $H'$ is also an $\varepsilon$-cut sparsifier of $H$.

The following theorem states that an $\varepsilon$-spectral sparsifier can be constructed using a single pass and $O(\varepsilon^{-2} n \log n)$ space in insertion-only streams.

**Theorem A.5** ([65]). *There exists a single-pass streaming algorithm for constructing an $\varepsilon$-spectral sparsifier of an undirected graph in insertion-only streams using $O(\varepsilon^{-2} n \log n)$ space. The algorithm succeeds with high probability.*

To construct spectral sparsifiers in dynamic streams, we use the following theorem.

**Theorem A.6** ([64]). *There exists a single-pass streaming algorithm for constructing an $\varepsilon$-spectral sparsifier of an undirected unweighted graph in dynamic streams using $\tilde{O}(\varepsilon^{-2} n)$ space. The algorithm succeeds with high probability.*

**Effective resistances.** Finally, we review the definition of effective resistances, which are used during the construction of spectral sparsifiers.

**Definition A.7** (Effective resistance). Given an undirected graph $G = (V, E, w)$ and a pair of vertices $u, v \in V$, the *effective resistance* between $u$ and $v$ in $G$ is defined as

$$R_G(u, v) := \max_{\boldsymbol{x} \in \mathbb{R}^{|V|}, \boldsymbol{x} \neq \boldsymbol{0}} \frac{(x_u - x_v)^2}{\boldsymbol{x}^\top L_G \boldsymbol{x}},$$

where $L_G$ is the Laplacian matrix of $G$.

We will use the following lower bound on the effective resistance.

**Lemma A.8** ([75]). *Let $G = (V, E)$ be an undirected unweighted graph. For any $u, v \in V$, we have*

$$R_G(u, v) \geq \frac{1}{2} \left( \frac{1}{\deg(u)} + \frac{1}{\deg(v)} \right).$$

## B    Technical overview

In this section, we provide a high-level overview of our techniques.

### B.1    Technical overview of our algorithms for complete graphs

Our algorithms for complete graphs (i.e., Algorithm 1 and Algorithm 8) rely on the influential PIVOT algorithm by Ailon et al. [5] and the LP rounding algorithm by Chawla et al. [32]. The PIVOT algorithm begins by selecting a random permutation $\pi$ over the vertices of the graph. It then iteratively forms clusters by choosing the vertex with the smallest rank according to $\pi$, along with its neighbors in the graph. Once a cluster is formed, it is removed from the graph. This process continues until all vertices have been assigned to clusters. The LP rounding algorithm first solves an LP corresponding to Correlation Clustering, and then applies a PIVOT-based algorithm, using the LP solution to form clusters.

Next, we describe our algorithms. The high-level idea is to incorporate the above LP rounding approach with the "truncated" PIVOT algorithms [22, 29], where our predictions correspond to a feasible LP solution in some sense. Specifically, for dynamic streams, we maintain a certain number of $\ell_0$-samplers during the stream and use them to derive a truncated subgraph at the end of the stream. Then we run the PIVOT algorithm and the LP rounding algorithm on the subgraph respectively and obtain two clusterings. Finally, we output the clustering with the lower cost. For insertion-only streams, we employ two different methods to store at most $k$ neighbors for each vertex during the stream, then run the PIVOT algorithm on the two stored subgraphs. We obtain two clusterings and output the one with the lower cost.

The analysis is non-trivial, even in insertion-only streams. We categorize all clusters into pivot clusters and singleton clusters, and analyze their costs respectively. Our key observation is that the truncated version of the LP rounding algorithm is equivalent to the algorithm that first samples a subgraph $G'$ according to the predictions and then runs the "truncated" PIVOT algorithms on $G'$. Our main technical contribution is to prove that 1) the cost of pivot clusters produced by the truncated version of the LP rounding algorithm is at most $2.06\beta$ times the cost of optimal solution (Lemma 4.5 and Lemma E.7); 2) the optimal solution on $G'$ does not differ from the optimal solution on the original graph $G$ by a lot (Lemma E.9). In this way, our algorithms can keep the space small while achieving an approximation ratio better than 3 under good prediction quality.

### B.2    Technical overview of our algorithms for general graphs

Our algorithm for general graphs (i.e., Algorithm 2) is inspired by the $O(\log |E^-|)$-approximation algorithm by Ahn et al. [4], which sparifies the positive subgraph $G^+$ to $H^+$, stores all negative edges $E^-$ during the stream, and applies the multiplicative weight update method to approximately solve a linear program. The resulting LP solution then guides an influential ball-growing procedure [31, 44] on the sparsifier $H^+$.

The high-level idea of our algorithm is to incorporate the above algorithm with our pairwise distance predictions, which, in a sense, form a feasible LP solution. Specifically, we also maintain a sparsifier $H^+$ for $G^+$ during the stream and, in the post-processing phase, perform ball-growing on $H^+$ using the predictions as distance metrics. Notably, we no longer need to store $E^-$ during the stream, leading to improved space complexity.

However, the approximation guarantee of this straightforward approach includes a $\log n$ term, whereas our goal is to replace it with a tighter $\log |E^-|$ term. This motivates us to further refine the algorithm. Specifically, during the stream, we maintain the sparsifier $H^+$ as before, while simultaneously storing the arriving negative edges and tracking their space usage. If at any point the space used to store negative edges exceeds $\tilde{O}(\varepsilon^{-2} n)$, we immediately stop storing them. After the stream ends (at which point the sparsifier is ready), we proceed with the ball-growing procedure as described above. Otherwise (i.e., if the space used by the negative edges never exceeds the threshold), we run the

**Algorithm 3** Offline version of Algorithm 1
***
**Input:** Graph $G^+ = (V, E^+)$, oracle access to a $\beta$-level predictor $\Pi$
**Output:** Clustering/Partition of $V$ into disjoint sets
 1: Pick a random permutation of vertices $\pi : V \to \{1, \dots, n\}$.
 2: Initially, all vertices are unclustered and interesting.
 3: A vertex $u$ marks itself uninteresting if $\pi_u \geq \tau_u$ where $\tau_u := \frac{c}{\varepsilon} \cdot \frac{n \log n}{\deg^+(u)}$. Here, $\varepsilon \in (0, 1/4)$
    and $c$ is a universal large constant.
 4: Let $G_{\text{store}}$ be the graph induced by the interesting vertices.
 5: $\mathcal{C}_1 \leftarrow \text{TRUNCATEDPIVOT}(G^+, G_{\text{store}}, \pi)$
 6: $\mathcal{C}_2 \leftarrow \text{TRUNCATEDPIVOTWITHPRED}(G^+, G_{\text{store}}, \pi, \Pi)$
 7: $i \leftarrow \arg\min_{i=1,2}\{\text{cost}_G(\mathcal{C}_i)\}$
 8: **return** $\mathcal{C}_i$

***

**Algorithm 4** TRUNCATEDPIVOT$(G^+, H, \pi)$
***
**Input:** Graph $G^+ = (V, E^+)$, induced subgraph $H = (V_H, E_H)$ where $V_H \subseteq V$ and $E_H \subseteq E^+$,
    permutation $\pi : V \to \{1, \dots, n\}$
**Output:** Clustering/Partition of $V$ into disjoint sets
 1: Let $U^{(1)} \leftarrow V_H$ be the set of unclustered vertices in $V_H$.
 2: Let $t \leftarrow 1$.
 3: **while** $U^{(t)} \neq \emptyset$ **do**
 4:     Let $u \in U^{(t)}$ be the vertex with the smallest rank.
 5:     Mark $u$ as a pivot. Initialize a new *pivot cluster* $S^{(t)} \leftarrow \{u\}$.
 6:     For each vertex $v \in U^{(t)}$ such that $(u, v) \in E_H$, add $v$ to $S^{(t)}$.
 7:     Remove all vertices clustered at this iteration from $U^{(t)}$.
 8:     $t \leftarrow t + 1$.
 9: Each vertex $u \in V \setminus V_H$ joins the cluster of pivot $v$ with the smallest rank, if $(u, v) \in E^+$ and
    $\pi_v < \tau_u$.
10: Each unclustered vertex $u \in V$ creates a *singleton cluster*.
11: **return** the final clustering $\mathcal{C}$, which contains all pivot clusters and singleton clusters

***

post-processing phase of the algorithm by Ahn et al. [4]. In this way, our algorithm keeps the space small while achieving a near $O(\log |E^-|)$-approximation under good prediction quality.

## C   Omitted pseudocodes of Section 4

In this section, we give the omitted pseudocodes of Section 4: Algorithm 3, Algorithm 4, Algorithm 5, Algorithm 6 and Algorithm 7.

## D   Omitted proofs of Section 4

### D.1   Proof of Theorem 1.3

**Space complexity.** We first analyze the space complexity of Algorithm 1. For each vertex $u \in V$, we mainly store its rank $\pi_u$, positive degree $\deg^+(u)$, and $10c \log n \cdot \sigma_u$ independent $\ell_0$-samplers. We have the following lemma which states the space requirement of $\ell_0$-samplers.

**Lemma D.1** ([22]). *The $\ell_0$-samplers used in Algorithm 1 require $O(\varepsilon^{-1} n \log^4 n)$ words of space.*

Furthermore, by Theorem A.6, the maintenance of an $\varepsilon$-spectral sparsifier in dynamic streams requires $\tilde{O}(\varepsilon^{-2} n)$ words of space. Therefore, the space complexity of Algorithm 1 is $\tilde{O}(\varepsilon^{-2} n)$ words.

**Approximation guarantee.** Next, we analyze the approximation ratio of Algorithm 1. We rely on the following lemma.

**Lemma D.2** (Lemma 2 in [22]). *The $\ell_0$-samplers allow us to recover the positive edges incident to all interesting vertices with high probability.*

Therefore, Algorithm 1 works with the same set of edges as Algorithm 3 in the clustering phase with high probability. This implies that both algorithms return the same clustering with the same

---

**Algorithm 5** TRUNCATEDPIVOTWITHPRED$(G^+, H, \pi, \Pi)$

---

**Input:** Graph $G^+ = (V, E^+)$, induced subgraph $H = (V_H, E_H)$ where $V_H \subseteq V$ and $E_H \subseteq E^+$, permutation $\pi : V \to \{1, \ldots, n\}$, oracle access to a $\beta$-level predictor $\Pi$
**Output:** Clustering/Partition of $V$ into disjoint sets
1: Let $U^{(1)} \leftarrow V_H$ be the set of unclustered vertices in $V_H$.
2: Let $t \leftarrow 1$.
3: For any $u, v \in V$, $d_{uv} = \Pi(u, v)$.
4: For any $u, v \in V$, define $p_{uv} := f(d_{uv})$.
5: **while** $U^{(t)} \neq \emptyset$ **do**
6:     Let $u \in U^{(t)}$ be the vertex with the smallest rank.
7:     Mark $u$ as a pivot. Initialize a new *pivot cluster* $S^{(t)} \leftarrow \{u\}$.
8:     For each vertex $v \in U^{(t)}$, add $v$ to $S^{(t)}$ with probability $(1 - p_{uv})$ independently.
9:     Remove all vertices clustered at this iteration from $U^{(t)}$.
10:     $t \leftarrow t + 1$.
11: Each vertex $u \in V \setminus V_H$ joins the cluster of pivot $v$ in the order of $\pi$ with probability $(1 - p_{uv})$ independently, if $\pi_v < \tau_u$.
12: Each unclustered vertex $u \in V$ creates a *singleton cluster*.
13: **return** the final clustering $\mathcal{C}$, which contains all pivot clusters and singleton clusters

---

**Algorithm 6** CKLPU-PIVOT$(G^+)$

---

**Input:** Graph $G^+ = (V, E^+)$
**Output:** Clustering/Partition of $V$ into disjoint sets
1: Pick a random permutation of vertices $\pi : V \to \{1, \ldots, n\}$.
2: Let $U^{(1)} \leftarrow V$ be the set of unclustered vertices.
3: **for** $t = 1, \ldots, n$ **do**
4:     Let $\ell_t \leftarrow \frac{c}{\varepsilon} \cdot \frac{n \log n}{t}$.
5:     Let $u \in V$ be the $t$-th vertex in $\pi$ (i.e., $t = \pi_u$).
6:     Each unclustered vertex $v$ with $\deg^+(v) \geq \ell_t$ creates a *singleton cluster*.
7:     **if** $u \in U^{(t)}$ **then**
8:         Mark $u$ as a pivot. Initialize a new *pivot cluster* $S^{(t)} \leftarrow \{u\}$.
9:         For each vertex $v \in N^+(u) \cap U^{(t)}$, add $v$ to $S^{(t)}$.
10:     Remove all vertices clustered at this iteration from $U^{(t)}$.
11: **return** the final clustering $\mathcal{C}$, which contains all pivot clusters and singleton clusters

---

probability. On the other hand, if the high probability event of Lemma D.2 does not happen, then Algorithm 1 produces a clustering of cost at most $O(n^2)$, which leads to an additive $1/\operatorname{poly}(n)$ term to the expected cost of Algorithm 1 compared to that of Algorithm 3. This preserves the approximation ratio if $\text{OPT} \neq 0$.

We also need the following lemma which shows that the estimate $\widetilde{\text{cost}}_G(\mathcal{C})$ well approximates the cost of any clustering $\mathcal{C}$ of $G$.

**Lemma D.3** ([18]). *Let $\varepsilon \in (0, 1)$. For any clustering $\mathcal{C}$ of $V$, the cost $\text{cost}_G(\mathcal{C})$ is approximated by the estimate $\widetilde{\text{cost}}_G(\mathcal{C}) := \sum_{C \in \mathcal{C}} \left( \frac{1}{2} \partial_{H^+}(C) + \binom{|C|}{2} - \frac{1}{2} \sum_{u \in C} \deg^+(u) \right)$ up to a multiplicative factor of $(1 \pm \varepsilon)$ with high probability.*

Therefore, Theorem 1.3 follows from Lemma 4.1, Lemma D.2 and Lemma D.3 by applying the union bound.

## D.2   Proof of Lemma 4.3

The proof is similar to that of Lemma 4.2. The proof idea is as follows: we first show that in both cases, the singleton clusters $V_{\sin}$ are the same (with the same probability). Then we show that the randomized pivot-based algorithm runs on the same subgraph $G^+[V \setminus V_{\sin}]$ (with the same probability) in both cases, therefore outputting the same pivot clusters (with the same probability).

Consider a vertex $u$ that is unclustered at the beginning of iteration $t$ ($\leq \pi_u$), and becomes a singleton cluster due to Line 8 of Algorithm PAIRWISEDISS. By definition, $t$ is the smallest integer such that

**Algorithm 7** PAIRWISEDISS($G^+, \Pi$)

**Input:** Graph $G^+ = (V, E^+)$, oracle access to a $\beta$-level predictor $\Pi$
**Output:** Clustering/Partition of $V$ into disjoint sets
1: Pick a random permutation of vertices $\pi : V \to \{1, \ldots, n\}$.
2: For any $u, v \in V$, $d_{uv} = \Pi(u, v)$.
3: For any $u, v \in V$, define $p_{uv} := f(d_{uv})$.
4: Let $U^{(1)} \leftarrow V$ be the set of unclustered vertices.
5: **for** $t = 1, \ldots, n$ **do**
6:     Let $\ell_t \leftarrow \frac{c}{\varepsilon} \cdot \frac{n \log n}{t}$.
7:     Let $u \in V$ be the $t$-th vertex in $\pi$ (i.e., $t = \pi_u$).
8:     Each unclustered vertex $v$ with $\deg^+(v) \geq \ell_t$ creates a *singleton cluster*.
9:     **if** $u \in U^{(t)}$ **then**
10:         Mark $u$ as a pivot. Initialize a new *pivot cluster* $S^{(t)} \leftarrow \{u\}$.
11:         For each vertex $v \in U^{(t)}$, add $v$ to $S^{(t)}$ with probability $(1 - p_{uv})$ independently.
12:     Remove all vertices clustered at this iteration from $U^{(t)}$.
13: **return** the final clustering $\mathcal{C}$, which contains all pivot clusters and singleton clusters

$\deg^+(u) \geq \frac{c}{\varepsilon} \cdot \frac{n \log n}{t}$ and hence $t = \lceil \tau_u \rceil$. Since $t \leq \pi_u$, we have $\deg^+(u) \geq \frac{c}{\varepsilon} \cdot \frac{n \log n}{\pi_u}$, which corresponds to $u$ becoming uninteresting in Algorithm 3. Since $u$ is in a singleton cluster, it did not join any pivot cluster, implying that for any vertex $v \neq u$, either (1) $\pi_v \geq t$, or (2) the event that $v$ is a pivot and $u$ joins the cluster of $v$ satisfying $\pi_v < t$ does not happen. This is equivalent to saying that the event that $u$ joins the cluster of pivot $v$ satisfying $\pi_v < \tau_u$ does not happen, since $\pi_v$ is an integer. By Line 11 of Algorithm TRUNCATEDPIVOTWITHPRED, $u$ creates a singleton cluster in Line 6 of Algorithm 3 (with the same probability) as well.

Now consider a vertex $u$ that creates a singleton cluster in Line 6 of Algorithm 3. Then $u$ must be marked uninteresting (implying $\pi_u \geq \tau_u$), and $u$ can neither be a pivot nor join the cluster of pivot $v$ satisfying $\pi_v < \tau_u$. By definition of $\tau_u$, iteration $\lceil \tau_u \rceil$ is the smallest iteration such that $\deg^+(u) \geq \frac{c}{\varepsilon} \cdot \frac{n \log n}{\lceil \tau_u \rceil}$. This implies that $u$ is unclustered at the beginning of iteration $\lceil \tau_u \rceil$ in Algorithm PAIRWISEDISS, and forms a singleton cluster in that iteration (with the same probability).

Since the vertices forming singleton clusters are the same in both cases (with the same probability), the subgraph induced by the remaining vertices $G^+[V \setminus V_{\text{sin}}]$ is the same (with the same probability). The same randomized pivot-based algorithm runs on $G^+[V \setminus V_{\text{sin}}]$ in both cases, which implies that the pivots will be the same (with the same probability). Finally, we observe that in both cases, a non-pivot vertex $u$ joins the cluster of pivot $v$ such that $\pi_v < \tau_u$ in the order of $\pi$ with probability $(1 - p_{uv})$ independently. Hence, the pivot clusters are the same (with the same probability).

### D.3   Proof of Corollary 4.6

The analysis in [22] shows that the cost of good edges can be charged to the pivot clusters. The following lemma shows that the cost of bad edges can also be related to the pivot clusters, allowing us to bound the overall clustering cost.

**Lemma D.4** ([22]). *Let $\varepsilon \in (0, 1/4)$. Let $P$ denote the cost of pivot clusters, and let $W$ denote the cost of the final clustering, then $\mathbb{E}[W] = \mathbb{E}[P + |E_{\text{bad}}|] \leq (1 + 4\varepsilon)\mathbb{E}[P] + \frac{1+4\varepsilon}{n^{\alpha-2}}$, where $\alpha := c/2 - 1 \gg 2$.*

Then Corollary 4.6 follows from Lemma 4.3, Lemma D.4 and Lemma 4.5.

### D.4   Proof of Lemma 4.1

Lemma 4.1 follows from Lemma 4.4 and Corollary 4.6. Note that in Lemma 4.4, we can substitute $\varepsilon' := 12\varepsilon$, where $\varepsilon$ can be arbitrarily small. If $\text{OPT} \geq 1$, then $\mathbb{E}[\text{cost}_G(\mathcal{C}_1)] \leq (3 + 12\varepsilon) \cdot \text{OPT}$, which gives a $(3 + \varepsilon')$-approximation in expectation. If $\text{OPT} = 0$, then $\mathbb{E}[\text{cost}_G(\mathcal{C}_1)] = 1/\text{poly}(n)$. Similarly, in Corollary 4.6, we can substitute $\varepsilon' := 8.24\beta\varepsilon$.

# E  Our algorithm for complete graphs in insertion-only streams

In this section, we propose an algorithm for complete graphs in insertion-only streams. This algorithm is different from our algorithm for dynamic streams (Algorithm 1), while achieving the same approximation guarantee with improved space complexity. It is also simpler and more practical than the existing 1.847-approximation algorithm [39], which is based on local search and requires enumerating a large number of subsets of a constant-size set, making it quite impractical.

**Theorem E.1.** *Let $\varepsilon \in (0,1)$ and $\beta \geq 1$. Given oracle access to a $\beta$-level predictor, there exists a single-pass streaming algorithm that, with high probability, achieves an expected $(\min\{2.06\beta, 3\} + \varepsilon)$-approximation for Correlation Clustering on complete graphs in insertion-only streams. The algorithm uses $O(\varepsilon^{-2} n \log n)$ words of space.*

## E.1  Overview

We first briefly describe a single-pass $(3 + \varepsilon)$-approximation streaming algorithm by Chakrabarty and Makarychev [29]. Initially, the algorithm adds a positive self-loop for each vertex and picks a random ordering $\pi : V \to \{1, \ldots, n\}$. The rank of $u$ is denoted as $\pi_u$. Then it scans the input stream. For each vertex, the algorithm stores its at most $k$ positive neighbors with lowest ranks, where $k$ is a constant. Subsequently, it runs the PIVOT algorithm [5] on the stored graph, where it picks pivots in the order of $\pi$. Finally, it puts unclustered vertices in singleton clusters.

Our main idea is to incorporate the above algorithm with the algorithm proposed by Chawla et al. [32]. Specifically, our algorithm employs two different methods to store at most $k$ neighbors of each vertex. The first method is the same as the algorithm proposed by Chakrabarty and Makarychev [29] and the second method is adapted from the work of Chawla et al. [32], which adds neighbors with probabilities determined by predictions of pairwise distances. Finally, we obtain two clusterings (denoted as $\mathcal{C}_1$ and $\mathcal{C}_2$) and output the one with the lower cost. Similar to Algorithm 1, here we also need to use the graph sparsification technique [65] to approximate the cost of a clustering.

## E.2  The algorithm

Recall that we have oracle access to a $\beta$-level predictor $\Pi$, which can predict the pairwise distance $d_{uv} \in [0, 1]$ between any two vertices $u$ and $v$ in $G$.

Based on the predictions, we propose a single-pass semi-streaming algorithm which works in insertion-only streams. The pseudocode is given in Algorithm 8. We first pick a random permutation of vertices $\pi : V \to \{1, \ldots, n\}$. For each vertex $u \in V$, we initialize two priority queues $A(u)$ and $B(u)$, each with a maximum size capped at $k$, where $k$ is a constant. Initially, we add $u$ to both queues. During the streaming phase, we employ two distinct methods to retain at most $k$ neighbors of each vertex. Specifically, for each edge $(u, v) \in E$ in the stream, if $(u, v)$ is a positive edge, we add $u$ to $A(v)$ and add $v$ to $A(u)$. Additionally, regardless of whether $(u, v)$ is positive or negative, we add $u$ to $B(v)$ with probability $(1 - p_{uv})$ and add $v$ to $B(u)$ with probability $(1 - p_{uv})$, where $p_{uv} = f(d_{uv})$ and $d_{uv} = \Pi(u, v)$. Note that if the size of any queue exceeds $k$, then we remove the vertex with the highest rank from the queue. That is, $A(u)$ maintains at most $k$ positive neighbors of $u$ with lowest ranks, while $B(u)$ contains at most $k$ neighbors (not necessarily positive) of $u$ with lowest ranks, the inclusion of which is probabilistic. Note that we define the rank of a vertex as its order in the permutation $\pi$, e.g., $\pi_u$ is the rank of $u$.

After the streaming phase, we run Algorithm 9 on the truncated graphs induced by both sets of priority queues, i.e., $\{A(u)\}_{u \in V}$ and $\{B(u)\}_{u \in V}$. Specifically, for each vertex $u$ picked in the order of $\pi$, we determine the cluster to which $u$ belongs. We try to find the vertex $v$ with the lowest rank in the queue of $u$, such that $v$ is a pivot or $v = u$. If such a vertex $v$ does not exist, then we mark $u$ as a singleton and place it in a singleton cluster. Otherwise, we assign $u$ to the cluster of $v$. In particular, if $v = u$, then we mark $u$ as a pivot. Finally, we obtain two clusterings, each corresponding to a set of priority queues. We output the clustering with the lower cost.

It is worth noting that in the final step, the cost of a clustering cannot be exactly calculated, as our streaming algorithm cannot store the entire graph. To overcome this challenge, we borrow the idea from the work of Behnezhad et al. [18] and utilize the graph sparsification technique [65] to estimate the the cost of a clustering. Specifically, during the streaming phase, we maintain an $\varepsilon$-spectral sparsifier $H^+$ for $G^+$ using the algorithm of Theorem A.5. We also maintain the

---

**Algorithm 8** An algorithm for complete graphs in insertion-only streams

---

**Input:** Complete graph $G = (V, E = E^+ \cup E^-)$ as an arbitrary-order stream of edges, oracle access to a $\beta$-level predictor $\Pi$, integer $k$
**Output:** Clustering/Partition of $V$ into disjoint sets
    ▷ **Pre-processing phase**
 1: Pick a random permutation of vertices $\pi : V \to \{1, \ldots, n\}$.
 2: For any $u, v \in V$, $d_{uv} = \Pi(u, v)$.
 3: For any $u, v \in V$, define $p_{uv} := f(d_{uv})$.
 4: **for** each vertex $u \in V$ **do**
 5:     Create a priority queue $A(u)$ with a maximum size of $k$ and initialize $A(u) \leftarrow \{u\}$.
 6:     Create a priority queue $B(u)$ with a maximum size of $k$ and initialize $B(u) \leftarrow \{u\}$.
 7:     $\deg^+(u) \leftarrow 0$
    ▷ **Streaming phase**
 8: **for** each edge $e = (u, v) \in E$ **do**
 9:     **if** $e = (u, v) \in E^+$ **then**
10:         Add $u$ to $A(v)$. Add $v$ to $A(u)$.
11:         **if** $|A(u)| > k$ (resp. $|A(v)| > k$) **then**
12:             Remove the vertex with the highest rank from $A(u)$ (resp. $A(v)$).
13:         $\deg^+(u) \leftarrow \deg^+(u) + 1, \deg^+(v) \leftarrow \deg^+(v) + 1$
14:     With probability $(1 - p_{uv})$, add $u$ to $B(v)$ and add $v$ to $B(u)$.
15:     **if** $|B(u)| > k$ (resp. $|B(v)| > k$) **then**
16:         Remove the vertex with the highest rank from $B(u)$ (resp. $B(v)$).
17: Maintain an $\varepsilon$-spectral sparsifier $H^+$ for $G^+$ using the algorithm of Theorem A.5.
    ▷ **Post-processing phase**
18: $\mathcal{C}_1 \leftarrow \text{CLUSTER}(V, \pi, \{A(u)\}_{u \in V})$
19: $\mathcal{C}_2 \leftarrow \text{CLUSTER}(V, \pi, \{B(u)\}_{u \in V})$
20: $\widetilde{\text{cost}}_G(\mathcal{C}_1) \leftarrow \sum_{C \in \mathcal{C}_1} (\frac{1}{2} \partial_{H^+}(C) + \binom{|C|}{2} - \frac{1}{2} \sum_{u \in C} \deg^+(u))$
21: $\widetilde{\text{cost}}_G(\mathcal{C}_2) \leftarrow \sum_{C \in \mathcal{C}_2} (\frac{1}{2} \partial_{H^+}(C) + \binom{|C|}{2} - \frac{1}{2} \sum_{u \in C} \deg^+(u))$
22: $i \leftarrow \arg\min_{i=1,2} \{\widetilde{\text{cost}}_G(\mathcal{C}_i)\}$.
23: **return** $\mathcal{C}_i$

---

**Algorithm 9** $\text{CLUSTER}(V, \pi, \{T(u)\}_{u \in V})$

---

**Input:** Vertex set $V$, permutation of vertices $\pi : V \to \{1, \ldots, n\}$, truncated neighbors of each vertex $\{T(u)\}_{u \in V}$
**Output:** Clustering/Partition of $V$ into disjoint sets
 1: **for** each unclustered vertex $u \in V$ chosen in the order of $\pi$ **do**
 2:     Find the vertex $v \in T(u)$ with the lowest rank such that $v$ is a pivot or $v = u$, i.e., $v \leftarrow \arg\min_{v \in T(u)} \{\pi_v : v \text{ is a pivot or } v = u\}$.
 3:     **if** such a vertex $v$ exists **then**
 4:         Put $u$ in the cluster of $v$.
 5:         **if** $v = u$ **then**
 6:             Mark $u$ as a *pivot*.
 7:     **else**
 8:         Put $u$ in a singleton cluster. Mark $u$ as a *singleton*.
 9: **return** the final clustering $\mathcal{C}$

---

positive degree $\deg^+(u)$ for each vertex $u$. Then we can approximate the cost of a clustering up to a $(1 \pm \varepsilon)$-multiplicative error with high probability, by the guarantee of Lemma D.3.

### E.3   Analysis of Algorithm 8

**Space complexity.** For each vertex $u \in V$, we mainly store its rank $\pi_u$, positive degree $\deg^+(u)$, and at most $2k$ vertices. As we will see, we set $k = O(1/\varepsilon)$. Furthermore, by Theorem A.5, the maintenance of an $\varepsilon$-spectral sparsifier in insertion-only streams requires $O(\varepsilon^{-2} n \log n)$ words of space. Therefore, the total space complexity of Algorithm 8 is $O(\varepsilon^{-2} n \log n)$ words.

---

**Algorithm 10** CM-PIVOT$(G, k)$

---

**Input:** Complete graph $G = (V, E = E^+ \cup E^-)$, integer $k$
**Output:** Partition of vertices into disjoint sets
1: Let $F^{(1)} \leftarrow V$ be the set of fresh vertices.
2: Let $U^{(1)} \leftarrow V$ be the set of unclustered vertices.
3: For each vertex $u \in V$, initialize a counter $K^{(1)}(u) \leftarrow 0$.
4: Let $t \leftarrow 1$.
5: **while** $F^{(t)} \neq \emptyset$ **do**
6:      Choose a vertex $w^{(t)} \in F^{(t)}$ uniformly at random.
7:      **if** $w^{(t)} \in U^{(t)}$ **then**
8:          Mark $w^{(t)}$ as a pivot. Initialize a new *pivot cluster* $S^{(t)} \leftarrow \{w^{(t)}\}$.
9:          For each vertex $v \in N^+(w^{(t)}) \cap U^{(t)}$, add $v$ to $S^{(t)}$.
10:      **else**
11:          For each vertex $v \in N^+(w^{(t)}) \cap U^{(t)}$, let $K^{(t+1)}(v) \leftarrow K^{(t)}(v) + 1$. Subsequently, all
      vertices $v$ with $K^{(t+1)}(v) = k$ are put into *singleton clusters*.
12:      Let $F^{(t+1)} \leftarrow F^{(t)} \setminus \{w^{(t)}\}$ and remove all vertices clustered at this iteration from $U^{(t)}$.
13:      Let $t \leftarrow t + 1$.
14: **return** the final clustering $\mathcal{C}$, which contains all pivot clusters and singleton clusters

---

**Correctness.** Since the final clustering returned by our algorithm is the one with the lower cost between the two on the truncated graphs, we begin by analyzing their costs. Similar to the analysis of Algorithm 1, for ease of analysis, we separately examine the approximation ratios of the corresponding offline versions (Algorithms CM-PIVOT and PAIRWISEDISS2) that equivalently output these two clusterings.

**Algorithm CM-PIVOT [29].** This algorithm proceeds in iterations. Let $F^{(t)}$ denote the set of fresh vertices and $U^{(t)}$ denote the set of unclustered vertices at the beginning of iteration $t$. Additionally, we maintain a counter $K^{(t)}(u)$ for each vertex $u \in V$. Initially, all the vertices are fresh and unclustered, with the counters set to 0. At iteration $t$, we pick a vertex $w^{(t)}$ from the set of fresh vertices $F^{(t)}$ uniformly at random. If $w^{(t)}$ is unclustered, then we mark it as a pivot and create a cluster $S^{(t)}$ containing $w^{(t)}$ and all of its unclustered positive neighbors. Otherwise, we increment the counters for all unclustered positive neighbors of $w^{(t)}$. Subsequently, vertices whose counters reach the value of $k$ are assigned to singleton clusters. At the end of iteration $t$, we remove $w^{(t)}$ from $F^{(t)}$ and remove all vertices clustered in this iteration from $U^{(t)}$. Then the algorithm proceeds to the next iteration. Finally, we output all pivot clusters and singleton clusters. The pseudocode is given in Algorithm 10.

**Algorithm PAIRWISEDISS2.** This algorithm has oracle access to a $\beta$-level predictor $\Pi : \binom{V}{2} \to [0, 1]$. This algorithm closely resembles Algorithm CM-PIVOT, differing in the following two aspects: (1) If $w^{(t)} \in U^{(t)}$, then we create a cluster $S^{(t)}$ containing $w^{(t)}$ and add all unclustered vertices $v$ to $S^{(t)}$ with probability $(1 - p_{vw^{(t)}})$ independently, where $p_{vw^{(t)}} = f(d_{vw^{(t)}})$ and $d_{vw^{(t)}} = \Pi(v, w^{(t)})$. (2) If $w^{(t)} \notin U^{(t)}$, we increment the counters for all unclustered vertices $v$ with probability $(1 - p_{vw^{(t)}})$. The pseudocode is given in Algorithm 11.

### E.3.1 Algorithm 8 as a combination of Algorithms CM-PIVOT and PAIRWISEDISS2

We define a permutation $\pi$ for Algorithms CM-PIVOT and PAIRWISEDISS2 as $\pi : w^{(t)} \mapsto t$, where $w^{(t)}$ is the vertex picked at iteration $t$ of Algorithms CM-PIVOT and PAIRWISEDISS2. Obviously, $\pi$ is a uniformly random permutation over $V$. Therefore, we can also view Algorithms CM-PIVOT and PAIRWISEDISS2 from an equivalent perspective: at the beginning of each iteration $t$, choose a vertex $w^{(t)}$ in the order of $\pi$. We have the following lemmas.

**Lemma E.2** (Lemma 2.1 in [29])**.** *If Algorithm 8 and Algorithm* CM-PIVOT *use the same permutation $\pi$, then Algorithm* CM-PIVOT *and Line 18 of Algorithm 8 output the same clustering of $V$.*

**Lemma E.3.** *If Algorithm 8 and Algorithm* PAIRWISEDISS2 *use the same permutation $\pi$ and predictions $\{d_{uv}\}_{u,v \in V}$, then Algorithm* PAIRWISEDISS2 *and Line 19 of Algorithm 8 output the same clustering of $V$ with the same probability.*

**Algorithm 11** PAIRWISEDISS2($G, \Pi, k$)

---

**Input:** Complete graph $G = (V, E = E^+ \cup E^-)$, oracle access to a $\beta$-level predictor $\Pi$, integer $k$
**Output:** Partition of vertices into disjoint sets
 1: Let $F^{(1)} \leftarrow V$ be the set of fresh vertices.
 2: Let $U^{(1)} \leftarrow V$ be the set of unclustered vertices.
 3: For each vertex $u \in V$, initialize a counter $K^{(1)}(u) \leftarrow 0$.
 4: For any $u, v \in V$, $d_{uv} = \Pi(u, v)$.
 5: For any $u, v \in V$, define $p_{uv} := f(d_{uv})$.
 6: Let $t \leftarrow 1$.
 7: **while** $F^{(t)} \neq \emptyset$ **do**
 8:     Choose a vertex $w^{(t)} \in F^{(t)}$ uniformly at random.
 9:     **if** $w^{(t)} \in U^{(t)}$ **then**
10:         Mark $w^{(t)}$ as a pivot. Initialize a new *pivot cluster* $S^{(t)} \leftarrow \{w^{(t)}\}$.
11:         For each vertex $v \in U^{(t)}$, add $v$ to $S^{(t)}$ with probability $(1 - p_{vw^{(t)}})$ independently.
12:     **else**
13:         For each vertex $v \in U^{(t)}$, let $K^{(t+1)}(v) \leftarrow K^{(t)}(v) + 1$ with probability $(1 - p_{vw^{(t)}})$
    independently. Subsequently, all vertices $v$ with $K^{(t+1)}(v) = k$ are put into *singleton clusters*.
14:     Let $F^{(t+1)} \leftarrow F^{(t)} \setminus \{w^{(t)}\}$ and remove all vertices clustered at this iteration from $U^{(t)}$.
15:     Let $t \leftarrow t + 1$.
16: **return** the final clustering $\mathcal{C}$, which contains all pivot clusters and singleton clusters

---

*Proof.* The proof is similar to that of Lemma E.2. Suppose that Algorithm 8 and Algorithm PAIR-WISEDISS2 use the same permutation $\pi$ and predictions $\{d_{uv}\}_{u,v \in V}$, we want to prove that for each vertex $u \in V$, with the same probability, in both clusterings returned by Algorithm PAIRWISEDISS2 and Line 19 of Algorithm 8, $u$ is either assigned to the same pivot, or $u$ is placed into a singleton cluster.

We prove by induction on the rank $\pi_u$. Suppose that all vertices $v$ with $\pi_v < \pi_u$ are clustered in the same way with the same probability. If $u$ is put into a singleton cluster in the clustering returned by Line 19 of Algorithm 8, then there must exist $k$ vertices added to $B(u)$ probabilistically, and their ranks are lower than $\pi_u$. None of the vertices in $B(u)$ are pivots. Since both algorithms use the same $\pi$ and $\{d_{uv}\}_{u,v \in V}$, in Algorithm PAIRWISEDISS2, these $k$ vertices will cause the counter of $u$ to increment $k$ times probabilistically. Therefore, $u$ is also placed in a singleton cluster in the clustering returned by Algorithm PAIRWISEDISS2. And vice versa.

In Algorithm 8, if there are any pivots in $B(u)$ (or $u$ itself), then $u$ will be assigned to the pivot with the lowest rank (denoted as $v$). We have $\pi_v \leq \pi_u$ and $v$ has been added to $B(u)$ probabilistically. In Algorithm PAIRWISEDISS2, with the same probability, $v$ is marked as a pivot and $u$ is added to the cluster of $v$. And vice versa.

Therefore, Algorithm PAIRWISEDISS2 and Line 19 of Algorithm 8 cluster $u$ in the same way with the same probability. □

### E.3.2    The approximation ratios of CM-PIVOT and PAIRWISEDISS2

In order to analyze the approximation ratio of Algorithm 8, it suffices to analyze Algorithms CM-PIVOT and PAIRWISEDISS2 respectively. We follow the analysis framework in the work of Chakrabarty and Makarychev [29]. We categorize all iterations into *pivot iterations* and *singleton iterations*. Both iterations create some clusters. Consider iteration $t$ of both algorithms. If $w^{(t)} \in U^{(t)}$, then iteration $t$ is a pivot iteration; otherwise, it is a singleton iteration. We say that an edge $(u, v)$ is *decided* at iteration $t$ if both $u$ and $v$ were not clustered at the beginning of iteration $t$ (i.e., $u, v \in U^{(t)}$) but at least one of them was clustered at iteration $t$. Once an edge $(u, v)$ is decided, we can determine whether it contributes to the cost of the algorithm (i.e., the number of disagreements). Specifically, if $(u, v) \in E^+$, then it contributes to the cost of the algorithm if exactly one of $u$ and $v$ is assigned to the newly created cluster $S^{(t)}$; if $(u, v) \in E^-$, then it contributes to the cost of the algorithm if both $u$ and $v$ are assigned to the newly created cluster $S^{(t)}$.

Let $E^{(t)}$ denote the set of decided edges at pivot iteration $t$. Specifically, $E^{(t)} = \{(u, v) \in E : u, v \in U^{(t)}; u \in S^{(t)} \text{ or } v \in S^{(t)}\}$. Let $P^{(t)}$ denote the cost of decided edges at pivot iteration $t$. We call the

clusters created in pivot iterations *pivot clusters*. Let $P$ denote the cost of all pivot clusters. Therefore, $P = \sum_{t \text{ is a pivot iteration}} P^{(t)}$. Let $S$ denote the cost of all singleton clusters. Therefore, the cost of the algorithm is equal to $P + S$.

**Analysis of Algorithm CM-PIVOT.** We have the following guarantees of Algorithm CM-PIVOT.

**Lemma E.4** ([29])**.** *Let $P_1$ denote the cost of pivot clusters returned by Algorithm* CM-PIVOT*, then* $\mathbb{E}[P_1] \leq 3 \cdot \mathrm{OPT}$*, where* $\mathrm{OPT}$ *is the cost of the optimal solution on $G$.*

**Lemma E.5** ([29])**.** *Let $S_1$ denote the cost of singleton clusters returned by Algorithm* CM-PIVOT*, then* $\mathbb{E}[S_1] \leq \frac{6}{k-1} \cdot \mathrm{OPT}$*.*

Therefore, we can bound the cost of the clustering returned by Line 18 of Algorithm 8.

**Lemma E.6.** *Let $P_1$ and $S_1$ denote the costs of pivot clusters and singleton clusters, respectively, returned by Algorithm* CM-PIVOT*. Let $\mathcal{C}_1$ denote the clustering returned by Line 18 of Algorithm 8. Then* $\mathbb{E}[\mathrm{cost}_G(\mathcal{C}_1)] = \mathbb{E}[P_1 + S_1] \leq (3 + \frac{6}{k-1}) \cdot \mathrm{OPT}$*.*

*Proof.* Lemma E.6 follows from Lemma E.2, Lemma E.4 and Lemma E.5. □

**Analysis of Algorithm PAIRWISEDISS2.** Next, we analyze the approximation ratio of Algorithm PAIRWISEDISS2. We first bound the cost of pivot clusters.

**Lemma E.7.** *Let $P_2$ denote the cost of pivot clusters returned by Algorithm* PAIRWISEDISS2*, then* $\mathbb{E}[P_2] \leq 2.06\beta \cdot \mathrm{OPT}$*.*

*Proof.* The key observation is that the pivot iterations in Algorithm PAIRWISEDISS2 are equivalent to the iterations of 2.06-approximation LP rounding algorithm by Chawla et al. [32]: given that $w^{(t)}$ is unclustered (i.e., $w^{(t)} \in U^{(t)}$), the conditional distribution of $w^{(t)}$ is uniformly distributed in $U^{(t)}$, and the cluster created during this iteration contains $w^{(t)}$ and all unclustered vertices $v$ added with probability $(1 - p_{vw^{(t)}})$. Therefore, we can directly apply the triangle-based analysis in the work of Chawla et al. [32]. Define $L := \sum_{(u,v)\in E^+} d_{uv} + \sum_{(u,v)\in E^-}(1 - d_{uv})$. Since the predictor is $\beta$-level, by Definition 3.1, we have that the predictions $\{d_{uv}\}_{u,v \in V}$ satisfy triangle inequality and $L \leq \beta \cdot \mathrm{OPT}$. It follows that for all pivot iterations $t$, $\mathbb{E}[P_2^{(t)}] \leq 2.06 \cdot \mathbb{E}[L^{(t)}]$, where $P_2^{(t)}$ is the cost induced by the pivot cluster created at iteration $t$, and $L^{(t)} := \sum_{(u,v)\in E^+\cap E^{(t)}} d_{uv} + \sum_{(u,v)\in E^-\cap E^{(t)}}(1 - d_{uv})$. By linearity of expectation, we have $\mathbb{E}[P_2] = \mathbb{E}[\sum_{t \text{ is a pivot iteration}} P_2^{(t)}] = \sum_{t \text{ is a pivot iteration}} \mathbb{E}[P_2^{(t)}] \leq 2.06 \cdot L \leq 2.06\beta \cdot \mathrm{OPT}$. □

Then we bound the cost of singleton clusters returned by Algorithm PAIRWISEDISS2, denoted as $S_2$. We highlight that this part is non-trivial. Different from the analysis in the work of Chakrabarty and Makarychev [29] which designs a potential function and shows that it is a submartingale, we consider an algorithm equivalent to Algorithm PAIRWISEDISS2. In this algorithm, we construct a random subgraph $G' := (V, E'^+ \cup E'^-)$ where each edge $(u, v) \in E$ is added to $E'^+$ with probability $(1 - p_{uv})$ and added to $E'^-$ with the remaining probability. Then we perform Algorithm CM-PIVOT on $G'$. In other words, we first preround the $\beta$-level predictions $\{d_{uv}\}_{u,v \in V}$ to obtain a new instance $G'$ and then run Algorithm CM-PIVOT on $G'$ where the positive edges are induced by the predictions. The pseudocode is given in Algorithm 12.

We first show the equivalence of Algorithm PAIRWISEDISS2 and Algorithm PAIRWISEDISS2WITHPREROUNDING.

**Lemma E.8.** *If Algorithm* PAIRWISEDISS2 *and Algorithm* PAIRWISEDISS2WITHPREROUNDING *use the same permutation $\pi$ and predictions $\{d_{uv}\}_{u,v \in V}$, then they produce the same clustering with the same probability.*

*Proof.* The randomness in both algorithms comes from two sources: (1) the uniformly random permutation $\pi$ on vertices and (2) the probability that each vertex $v$ adjacent to $w^{(t)}$ will join the cluster of $w^{(t)}$ or increment its counter. The main difference between the two algorithms lies in the order in which the two sources of randomness are revealed: Algorithm PAIRWISEDISS2 can be viewed as choosing $\pi$ at the beginning and then performing iterations, where the randomness of all edges incident

---

**Algorithm 12** PAIRWISEDISS2WITHPREROUNDING($G, \Pi, k$)

---

**Input:** Complete graph $G = (V, E = E^+ \cup E^-)$, oracle access to a $\beta$-level predictor $\Pi$, integer $k$
**Output:** Partition of vertices into disjoint sets
  1: For any $u, v \in V$, $d_{uv} = \Pi(u, v)$.
  2: For any $u, v \in V$, define $p_{uv} := f(d_{uv})$.
  3: $E'^+ \leftarrow \emptyset$.
  4: **for** each edge $(u, v) \in E$ such that $p_{uv} < 1$ **do**
  5:     add $(u, v)$ to $E'^+$ with probability $(1 - p_{uv})$.
  6: $E'^- \leftarrow E \setminus E'^+$
  7: $\mathcal{C} \leftarrow$ CM-PIVOT($G' := (V, E'^+ \cup E'^-), k$)
  8: **return** $\mathcal{C}$

---

to $w^{(t)}$ is revealed after $w^{(t)}$ is chosen. In contrast, Algorithm PAIRWISEDISS2WITHPREROUNDING reveals the randomness of edges at the beginning, uses this information to construct a new instance, and then performs Algorithm CM-PIVOT on the new instance, where the randomness for $\pi$ is revealed. Note that the order of randomness does not affect the output. Therefore, if both algorithms use the same $\pi$ and $\{d_{uv}\}_{u,v \in V}$, then they will output the same clustering with the same probability. $\qquad\square$

Therefore, we can directly apply Lemma E.5 to $G'$. To this end, we first show that $G'$ still well preserves the Correlation Clustering structure of $G$, by showing that the optimal solution on $G'$ does not differ from the optimal solution on $G$ by a lot.

**Lemma E.9.** $\mathbb{E}[\text{OPT}'] \leq (2\beta + 1) \cdot \text{OPT}$, *where* OPT *and* OPT' *are the costs of the optimal solutions on $G$ and $G'$, respectively.*

*Proof.* Let $\mathcal{C}^*$ be the optimal clustering on $G$ with cost OPT. For any $u, v \in V$, let $x^*_{uv} \in \{0, 1\}$ indicate whether $u$ and $v$ are in the same cluster or not in $\mathcal{C}^*$. Specifically, if $u$ and $v$ are in the same cluster in $\mathcal{C}^*$, then $x^*_{uv} = 0$; otherwise, $x^*_{uv} = 1$. Let $\mathcal{C}'^*$ be the optimal clustering on $G'$ with cost OPT'. Then we have

$$\mathbb{E}[\text{OPT}'] = \mathbb{E}[\text{cost}_{G'}(\mathcal{C}'^*)] \leq \mathbb{E}[\text{cost}_{G'}(\mathcal{C}^*)]$$

$$= \sum_{(u,v) \in E^+} [x^*_{uv}(1 - p_{uv}) + (1 - x^*_{uv})p_{uv}] + \sum_{(u,v) \in E^-} [x^*_{uv}(1 - p_{uv}) + (1 - x^*_{uv})p_{uv}]$$

$$= \sum_{(u,v) \in E^+} x^*_{uv} + \sum_{(u,v) \in E^-} (1 - x^*_{uv}) + \sum_{(u,v) \in E^+} p_{uv}(1 - 2x^*_{uv}) + \sum_{(u,v) \in E^-} (1 - p_{uv})(2x^*_{uv} - 1)$$

$$\leq \text{OPT} + \sum_{(u,v) \in E^+} p_{uv} + \sum_{(u,v) \in E^-} (1 - p_{uv})$$

$$\leq \text{OPT} + \sum_{(u,v) \in E^+} 2d_{uv} + \sum_{(u,v) \in E^-} (1 - d_{uv}) \leq (1 + 2\beta) \cdot \text{OPT},$$

where the first step follows from $\text{cost}_{G'}(\mathcal{C}'^*) = \text{OPT}'$, the second step follows from that $\mathcal{C}'^*$ is the optimal clustering on $G'$, the third step follows from our construction of $G'$, the fifth step follows from $\sum_{(u,v) \in E^+} x^*_{uv} + \sum_{(u,v) \in E^-} (1 - x^*_{uv}) = \text{OPT}$ and $\sum_{(u,v) \in E^+} p_{uv}(1 - 2x^*_{uv}) + \sum_{(u,v) \in E^-} (1 - p_{uv})(2x^*_{uv} - 1) \leq \sum_{(u,v) \in E^+} p_{uv} + \sum_{(u,v) \in E^-} (1 - p_{uv})$ since $1 - 2x^*_{uv} \in \{-1, 1\}$, the sixth step follows from our choice for $p_{uv}$, and the last step follows from $\sum_{(u,v) \in E^+} 2d_{uv} + \sum_{(u,v) \in E^-} (1 - d_{uv}) \leq 2(\sum_{(u,v) \in E^+} d_{uv} + \sum_{(u,v) \in E^-} (1 - d_{uv})) \leq 2\beta \cdot \text{OPT}$. $\qquad\square$

Now we are ready to bound the cost of singleton clusters and, consequently, the final clustering returned by Algorithm PAIRWISEDISS2.

**Lemma E.10.** *Let $S_2$ denote the cost of singleton clusters returned by Algorithm PAIRWISEDISS2, then $\mathbb{E}[S_2] \leq \frac{6(2\beta+1)}{k-1} \cdot \text{OPT}$.*

*Proof.* By Lemma E.5, Lemma E.8 and Lemma E.9, we have $\mathbb{E}[S_2] \leq \frac{6}{k-1} \cdot \mathbb{E}[\text{OPT}'] \leq \frac{6(2\beta+1)}{k-1} \cdot \text{OPT}$. $\qquad\square$

**Corollary E.11.** *Let $\mathcal{C}_2$ denote the clustering returned by Line 19 of Algorithm 8, then* $\mathbb{E}[\mathrm{cost}_G(\mathcal{C}_2)] = \mathbb{E}[P_2 + S_2] \leq (2.06\beta + \frac{6(2\beta+1)}{k-1}) \cdot \mathrm{OPT}.$

*Proof.* Corollary E.11 follows from Lemma E.3, Lemma E.7 and Lemma E.10. □

*Proof of Theorem E.1.* Theorem E.1 follows from Lemma E.6, Corollary E.11 and Lemma D.3. □

**Remark.** The reason our sampling-based approach works is mainly due to the fact that the rounding algorithm by Chawla et al. [32] is equivalent to the algorithm that first samples a subgraph $G'$ according to the prediction oracle and then runs the PIVOT algorithm on $G'$. Therefore, if a Correlation Clustering algorithm $\mathcal{A}$ has a similar feature, i.e., can be viewed as a procedure that first obtains a core of the original graph (by using LP or other methods), and then applies the PIVOT algorithm on the core, then we can get roughly the same approximation ratio as $\mathcal{A}$.

# F Omitted details of Section 5

## F.1 Proof of Lemma 5.3

The positive cost of $\mathcal{C}_2$ on $H$ is

$$
\begin{aligned}
\mathrm{cost}_H^+(\mathcal{C}_2) = \mathrm{cost}_{H^+}(\mathcal{C}_2) &= \frac{1}{2} \sum_{C \in \mathcal{C}_2} \partial_{H^+}(C) \\
&\leq \frac{3}{2} \ln(n+1) \sum_{C \in \mathcal{C}_2} \mathrm{vol}_{H^+}(C) \\
&\leq \frac{3}{2} \ln(n+1) \left( \sum_{(u,v) \in E_H^+} w'_{uv} d_{uv} + \sum_{C \in \mathcal{C}_2} \frac{V^*}{n} \right) \\
&\leq 3 \ln(n+1) \cdot \sum_{(u,v) \in E_H^+} w'_{uv} d_{uv},
\end{aligned}
$$

where the second-to-last step follows from the triangle inequality, and the final step uses the fact that $V^* = \sum_{(u,v) \in E_H^+} w'_{uv} d_{uv}$ and $|\mathcal{C}_2| \leq n$.

## F.2 Proof of Lemma 5.4

Since

$$
\sum_{(u,v) \in E^-} (1 - d_{uv}) \geq \sum_{C \in \mathcal{C}_2} \sum_{(u,v) \in E^-: u,v \in C} (1 - d_{uv}) \geq \sum_{C \in \mathcal{C}_2} \sum_{(u,v) \in E^-: u,v \in C} \left(1 - \frac{2}{3}\right)
$$

$$
= \frac{1}{3} \sum_{C \in \mathcal{C}_2} |(u,v) \in E^- : u,v \in C|,
$$

where the second step follows from $d_{uv} \leq \frac{2}{3}$ by triangle inequality, we have

$$
\mathrm{cost}_H^-(\mathcal{C}_2) = \sum_{C \in \mathcal{C}_2} |(u,v) \in E^- : u,v \in C| \leq 3 \sum_{(u,v) \in E^-} (1 - d_{uv}).
$$

## F.3 Proof of Lemma 5.5

To analyze the cost of $\mathcal{C}_2$ on the original graph $G$, we first show that the positive cost of $\mathcal{C}_2$ on $G$ is close to that on $H$. Since $H^+$ is an $\varepsilon$-spectral sparsifier of $G^+$, we have $\mathrm{cost}_G^+(\mathcal{C}_2) = \frac{1}{2} \sum_{C \in \mathcal{C}_2} \partial_{G^+}(C) \leq \frac{1}{2} \sum_{C \in \mathcal{C}_2} \frac{1}{1-\varepsilon} \partial_{H^+}(C) = \frac{1}{1-\varepsilon} \mathrm{cost}_H^+(\mathcal{C}_2)$. Therefore, by Lemma 5.3 and Lemma 5.4, the cost of $\mathcal{C}_2$ on $G$ is

$$
\mathrm{cost}_G(\mathcal{C}_2) = \mathrm{cost}_G^+(\mathcal{C}_2) + \mathrm{cost}_G^-(\mathcal{C}_2) = \mathrm{cost}_G^+(\mathcal{C}_2) + \mathrm{cost}_H^-(\mathcal{C}_2) \leq \frac{1}{1-\varepsilon} \mathrm{cost}_H^+(\mathcal{C}_2) + \mathrm{cost}_H^-(\mathcal{C}_2)
$$

$$\leq \frac{3\ln(n+1)}{1-\varepsilon} \cdot \sum_{(u,v)\in E_H^+} w'_{uv}d_{uv} + 3 \sum_{(u,v)\in E^-} (1-d_{uv})$$

$$\leq (3+4\varepsilon)\ln(n+1) \cdot \sum_{(u,v)\in E_H^+} w'_{uv}d_{uv} + 3 \sum_{(u,v)\in E^-} (1-d_{uv})$$

$$\leq (3+4\varepsilon)\ln(n+1) \cdot \left( \sum_{(u,v)\in E_H^+} w'_{uv}d_{uv} + \sum_{(u,v)\in E^-} (1-d_{uv}) \right) = O(\beta \log |E^-|) \cdot \mathrm{OPT},$$

where the second step follows from $\mathrm{cost}_G^-(\mathcal{C}_2) = \mathrm{cost}_H^-(\mathcal{C}_2)$, the third-to-last step uses the inequality $\frac{3}{1-\varepsilon} < 3+4\varepsilon$ for any $\varepsilon \in (0, 1/4)$, and the last step follows from the definition of the adapted $\beta$-level predictor and the fact that $|E^-| \geq n$.

### F.4 Results under bounded degree graphs

If the input graph has bounded degree, then the adapted $\beta$-level predictor can be relaxed to the standard $\beta$-level predictor.

**Corollary F.1.** *Let $\varepsilon \in (0, 1/2)$ and $d, \beta \geq 1$. Given oracle access to a $\beta$-level predictor, there exists a single-pass streaming algorithm that, with high probability, achieves an $O(\log |E^-| + \beta d)$-approximation for Correlation Clustering on general graphs with maximum degree $d$ in dynamic streams. The algorithm uses $\tilde{O}(\varepsilon^{-2}n)$ words of space.*

*Proof.* Recall that the $\varepsilon$-spectral sparsifier $H^+$ is constructed using the algorithm of Theorem A.6, which is based on effective resistance-based sampling [64]. Specifically, each edge $(u, v)$ in the original graph $G^+$ is sampled with probability $p_{uv} \geq \frac{C\ln n}{\varepsilon^2} \cdot R_G(u, v)$, where $C$ is a sufficiently large constant. For any edge $(u, v) \in E_H^+$, its weight is assigned by $w'_{uv} = \frac{1}{p_{uv}} \leq \frac{\varepsilon^2}{C\ln n \cdot R_G(u,v)}$. Since the maximum degree of $G$ is $d$, by Lemma A.8, we have $R_G(u, v) \geq \frac{1}{2}(\frac{1}{\deg(u)} + \frac{1}{\deg(v)}) \geq \frac{1}{d}$ for all $u, v \in V$. It follows that $w'_{uv} \leq \frac{\varepsilon^2 d}{C\ln n}$ for all $(u, v) \in E_H^+$.

Then it suffices to follow the proof of Lemma 5.5 to obtain

$$\mathrm{cost}_G(\mathcal{C}_2) \leq \frac{3\ln(n+1)}{1-\varepsilon} \cdot \sum_{(u,v)\in E_H^+} w'_{uv}d_{uv} + 3 \sum_{(u,v)\in E^-} (1-d_{uv})$$

$$\leq \frac{3\varepsilon^2 d\ln(n+1)}{(1-\varepsilon)C\ln n} \cdot \sum_{(u,v)\in E_H^+} d_{uv} + 3 \sum_{(u,v)\in E^-} (1-d_{uv})$$

$$\leq \max\left\{ \frac{(1+\varepsilon)d}{C}\left(1 + \frac{1}{n\ln n}\right), 3 \right\} \cdot \left( \sum_{(u,v)\in E^+} d_{uv} + \sum_{(u,v)\in E^-} (1-d_{uv}) \right)$$

$$= O(\beta d) \cdot \mathrm{OPT},$$

where the third step uses the fact that $\frac{3\varepsilon^2}{1-\varepsilon} \leq 1+\varepsilon$ for any $\varepsilon \in (0, 1/2)$, that $\frac{\ln(n+1)}{\ln n} \leq 1 + \frac{1}{n\ln n}$ for sufficiently large $n$, and that $E_H^+ \subseteq E^+$; and the last step follows from the definition of the $\beta$-level predictor (Definition 3.1).

Combining Lemma 5.2 with the above analysis yields Corollary F.1. $\qquad\square$

## G Additional experiments

In this section, we provide detailed descriptions of the datasets and predictors used in the experiments. Additionally, we present further experimental results.

### G.1 Detailed descriptions of datasets

In this subsection, we give a detailed description of the real-world datasets used in our experiments. Recall that we use EMAILCORE [70, 94], FACEBOOK [78], LASTFM [84], and DBLP [93] from the

Table 3: Statistics of real-world datasets. Here, $n$ and $m$ denote the number of vertices and edges in the original datasets.

| Datasets | $n$ | $m$ |
|---|---|---|
| EMAILCORE | 1 005 | 25 571 |
| FACEBOOK | 4 039 | 88 324 |
| LASTFM | 7 624 | 27 806 |
| DBLP | 317 080 | 1 049 866 |

Stanford Large Network Dataset Collection [71]. We provide basic statistics about these datasets in Table 3.

EMAILCORE is a directed network with 1 005 vertices and 25 571 edges. This network is constructed based on email exchange data from a large European research institution. Each vertex represents a person in the institution. There is a directed edge $(u, v)$ in the network if person $u$ has sent at least one email to person $v$.

FACEBOOK is an undirected network with 4 039 vertices and 88 324 edges. This network consists of friend lists of users from Facebook. Each vertex represents a user in Facebook. There is an undirected edge $(u, v)$ in the network if $u$ and $v$ are friends. Due to the computational bottleneck of solving the LP, we only use its three ego-networks: FB 0 ($n = 333, m = 5 038$), FB 414 ($n = 150, m = 3 386$), FB 3980 ($n = 52, m = 292$).

LASTFM is an undirected network with 7 624 vertices and 27 806 edges. This network is a social network of LastFM users, collected from the public API. Each vertex represents a LastFM user from an Asian country. There is an undirected edge $(u, v)$ in the network if $u$ and $v$ are mutual followers.

DBLP is an undirected co-authorship network with 317 080 vertices and 1 049 866 edges. Each vertex represents an author. There is an undirected edge $(u, v)$ in the network if $u$ and $v$ publish at least one paper together. Ground-truth communities are defined based on publication venues: authors who have published in the same journal or conference belong to the same community. For our experiments, we use a sampled subgraph consisting of 10 000 vertices.

**Remark.** We treat the original edges in the datasets as positive edges and non-edges as negative implicitly. (For datasets used in experiments where binary classifiers are employed as predictors, the interpretation of positive and negative edges differs slightly. See Appendix G.2 for details.) For directed networks, we convert all directed edges into undirected edges. *We highlight that since we consider labeled complete graphs in the experiments, the number of edges scales quadratically w.r.t. the number of vertices, which leads to non-trivial instance sizes.*

### G.2 Detailed descriptions of predictors

**Noisy predictor.** We use this predictor for datasets with available optimal clusterings. We form this predictor by performing perturbations on optimal clusterings. Specifically, for any two vertices $u, v \in V$, if $u$ and $v$ are in different clusters in the optimal clustering, then we set the prediction $d_{uv}$ to be $1 - \varepsilon_0$, otherwise $\varepsilon_0$, where $\varepsilon_0 \in (0, 0.5)$. For synthetic datasets with $p > 0.9$, we can assume that the ground truths are also optimal solutions. For other datasets, we use the powerful LP solver Gurobi [55] to get the optimal clusterings.

**Spectral embedding.** We use this predictor for EMAILCORE and LASTFM. It first maps all the vertices to a $d$-dimensional Euclidean space using the graph Laplacian, then clusters all the vertices based on their embeddings. For any two vertices $u, v \in V$, we form the prediction $d_{uv}$ to be $1 - \frac{\langle \boldsymbol{x}_u, \boldsymbol{x}_v \rangle}{\|\boldsymbol{x}_u\| \|\boldsymbol{x}_v\|}$, where $\boldsymbol{x}_u, \boldsymbol{x}_v \in \mathbb{R}^d$ are spectral embeddings of $u$ and $v$, and $\langle \boldsymbol{x}_u, \boldsymbol{x}_v \rangle$ is the dot product of $\boldsymbol{x}_u$ and $\boldsymbol{x}_v$. Note that a larger $d$ indicates a higher-quality predictor.

**Binary classifier.** We use this predictor for datasets with available ground-truth communities. This predictor is constructed by training a binary classifier (based on an MLP model) to predict whether two vertices belong to the same cluster using node features. In this setting, the goal of Correlation Clustering aligns with that of community detection by treating edges between two vertices in the same

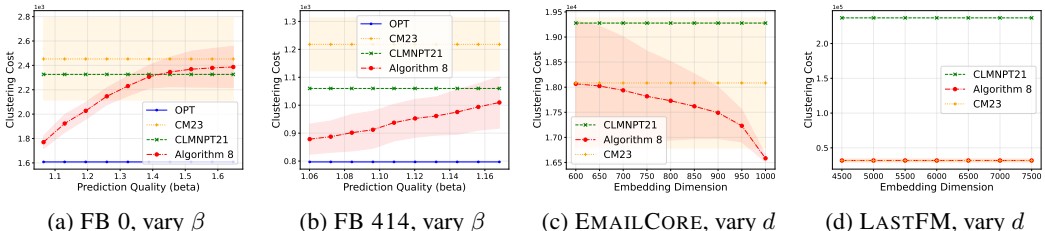

| (a) FB 0, vary $\beta$ | (b) FB 414, vary $\beta$ | (c) EMAILCORE, vary $d$ | (d) LASTFM, vary $d$ |

Figure 3: Performance of Algorithm 8 on real-world datasets. (a)–(b) show the effect of prediction quality $\beta$ on two FACEBOOK subgraphs, where we use noisy predictors. (c)–(d) examine the effect of the dimension $d$ of spectral embeddings on EMAILCORE and LASTFM, where we use spectral embedding as the predictor. We set $k = 25$ for (a), $k = 15$ for (b), $k = 10$ for (c), and $k = 50$ for (d).

Table 4: Clustering costs ($\times$ 1e3) of Algorithm 8 with binary classifiers as predictors, compared to its non-learning counterpart. We set parameter $k = 75$ across all datasets. The reported values are averaged over 5 runs.

| Dataset / Algorithm | SBM ($n = 1200$) | SBM ($n = 2400$) | SBM ($n = 3600$) | DBLP |
|---|---|---|---|---|
| CM23 | 99.3 | 385.7 | 901.6 | 236.4 |
| **Algorithm 8** | 35.9 | 155.3 | 324.9 | 214.9 |

(ground-truth) community as positive edges and edges between two vertices in different communities as negative edges. The predictions provided by the binary classifier (i.e., binary values in $\{0, 1\}$) are then used as the pairwise distances $d_{uv}$ in our algorithms.

## G.3 Additional results

### G.3.1 Performance of Algorithm 8 on real-world datasets

In this subsection, we present the results of our algorithm in insertion-only streams (Algorithm 8) on real-world datasets, as shown in Figure 3. The results show that under good prediction quality, Algorithm 8 consistently outperforms other baselines across all datasets used. For example, in Figure 3(a), when $\beta \approx 1.2$, the average cost of Algorithm 8 is $13\%$ lower than that of CLMNPT21 and $17\%$ lower than that of CKLPU24. Besides, in Figure 3(c), Algorithm 8 reduces the clustering cost by up to $14\%$ compared to CLMNPT21. Even if the prediction quality is poor, Algorithm 8 does not perform worse than CM23 and achieves comparable performance to CLMNPT21 (on FACEBOOK subgraphs).

### G.3.2 Performance of Algorithm 8 using binary classifiers as predictors

In this subsection, we present the results of Algorithm 8 using binary classifiers as predictors, as shown in Table 4. These experiments are performed on three SBM graphs with parameter $p = 0.95$ and varying sizes, as well as the DBLP dataset (a sampled subgraph with $10\,000$ vertices). The results demonstrate that Algorithm 8 consistently outperforms its non-learning counterpart across all datasets. For instance, on the SBM graphs with $1\,200$ and $3\,600$ vertices, Algorithm 8 achieves a $64\%$ reduction in clustering cost compared to CM23.

### G.3.3 Running time of our algorithms

In this subsection, we present the running time of our algorithms for complete graphs on three FACE-BOOK subgraphs, compared to their non-learning counterparts, as shown in Table 5 (Algorithm 1) and Table 6 (Algorithm 8). The results show that our learning-augmented algorithms do not introduce significant time overheads. The slight increase in running time is due to the additional steps of querying the oracles and calculating the costs of two clusterings. These steps are both reasonable and acceptable. Moreover, in the streaming setting, space efficiency is typically the primary focus.

Table 5: Running time (ms) of Algorithm 1 (for dynamic streams) on FACEBOOK subgraphs, compared to its non-learning counterpart. For FB 0, we set $\beta = 1.19$. For FB 414, we set $\beta = 1.12$. For FB 3980, we set $\beta = 1.19$.

| Algorithm ⟍ Dataset | FB 0 | FB 414 | FB 3980 |
|---|---|---|---|
| CKLPU24 | 1 738.16 | 165.55 | 7.32 |
| **Algorithm 1** | 1 639.22 | 163.35 | 7.69 |

Table 6: Running time (ms) of Algorithm 8 (for insertion-only streams) on FACEBOOK subgraphs, compared to its non-learning counterpart. For FB 0, we set $\beta = 1.19$. For FB 414, we set $\beta = 1.12$. For FB 3980, we set $\beta = 1.19$.

| Algorithm ⟍ Dataset | FB 0 | FB 414 | FB 3980 |
|---|---|---|---|
| CM23 | 30.65 | 6.67 | 0.97 |
| **Algorithm 8** | 81.31 | 16.58 | 2.12 |

