# OpenReview forum: "Learning-Augmented Streaming Algorithms for Correlation Clustering"
_NeurIPS.cc/2025/Conference — NeurIPS 2025 poster_

### Official Review · Reviewer_S28r · 2025-06-05

**Clarity:** 4
**Significance:** 3
**Originality:** 2
**Rating:** 4
**Confidence:** 4

**Summary:**

This paper tackles the problem of correlation clustering in the streaming setting (both insertion-only and dynamic streams) and proposes to augment classical streaming algorithms with predictions to improve performance. In correlation clustering, we receive a stream of edges (each labeled positive or negative) of a graph and aim to partition the vertices into clusters to minimize disagreements (positive edges across clusters plus negative edges within clusters). The authors introduce the first learning-augmented streaming algorithms for this problem. Their algorithms assume access to a predictor that provides pairwise “dissimilarity” scores between vertices (intended to estimate whether two vertices should be separated or not in the optimal clustering). Using these predictions, the paper’s algorithms achieve an approximation ratio better than 3 for clustering on complete graphs, which is the first time an approximation factor below 3 has been attained in the single-pass dynamic streaming model. For general (not necessarily complete) graphs, the paper presents a streaming algorithm that attains an $O(\beta \log |E^-|)$ approximation (here $|E^-|$ is the number of negative edges and $\beta$ measures prediction error) using only $\tilde O(n)$ space, improving the memory usage over the prior $O(|E^-|)$ space bound without predictions. The methods build upon and simplify prior techniques (notably the Pivot-based algorithms by Ahn et al. 2015 and Cambus et al. 2024) by incorporating predicted pairwise distances into the clustering decisions. The paper includes experiments on synthetic and real datasets, which indicate that the proposed learning-augmented algorithms outperform baseline streaming algorithms that do not use predictions.

**Questions:**

1. The theoretical model assumes an oracle predictor with a known $\beta$-level of quality. How do the authors envision obtaining such a predictor in practice? For instance, could it be trained on smaller samples or past instances of similar clustering problems, and how sensitive is the algorithm’s performance to mis-specification of $\beta$?

2. The paper mentions the $1.847$-approximation insertion-only streaming algorithm by Cohen-Addad et al. (STOC 2024), noting that the new approach is much simpler and works in the dynamic setting. Could the authors elaborate on the performance gap here? For example, if we restrict to insertion-only streams, does the learning-augmented algorithm attain a similar 2.06 approximation, or could it be improved further? A brief discussion on whether the learning-augmented approach might close the gap to $1.847$ in insertion-only scenarios (or why that might be difficult) would be insightful.

3. The general graph result uses an adapted $\beta$-level predictor. Is this essentially the same notion of predictor quality as in complete graphs, or does it require a different adaptation (e.g., predicting distances in a sparsified graph or dealing with multicut structure)? It would be helpful if the paper clarified what adjustments are needed for the predictor in general graphs and whether obtaining such an adapted predictor is harder or easier than in the complete graph case.

4. In practice, do the authors observe any memory overhead due to the learning-augmented approach (like needing to store predictor information)? If the space usage was tightened or if less space is available, how would the algorithm degrade?

5. Limited Novelty - The algorithm is essentially formed by the truncation during the stream and the post-processing part. For the streaming phase, the idea is mainly from CKLPU [SODA’24] (cf. CM [NeurIPS’23]), and I do not see anything that is substantially novel. Am I understanding it correctly, or am I missing something? Also, the approximation factor is somewhat weak; to outperform the pivot-based approximation, there has to be a value of $\beta$ where you need predictions with very high quality, correct?

**Ethical Concerns:**

["NO or VERY MINOR ethics concerns only"]

**Final Justification:**

All my questions have been answered. After reading the comments from other reviewers and the comprehensive responses from the authors, I think there is no need to update my score. I keep my score to 4 and I incline to accept the paper.

**Limitations:**

The paper does acknowledge its limitations. The authors clearly state all necessary assumptions (e.g., the need for an oracle predictor and its quality parameter $\beta$) and ensure the reader understands in which scenarios the theoretical guarantees hold. They also empirically test robustness by varying parameters like dataset size to see how the algorithm copes. The limitations I see are the following

1. The focus on correlation clustering on cliques is primarily a theoretical problem. Existing streaming algorithms already offer better guarantees for this specific case.

2. The novelty is somewhat limited, as the algorithm builds upon existing techniques and incorporates a learning component which does not mean that the novelty is significant. I again acknowledge that coming up with mathematical proofs and making the proof go through is not trivial.

3. Unnatural Oracle: This is a significant drawback in the paper. The ML oracle used by the algorithm seems somewhat artificial, and its practical implementation may be challenging. On a very high-level intuition, the way I see it is "Suppose we assume that we can predict similarities that could be used to obtain a good approximation of the optimal solution, then we have an approximation algorithm of that quality." Is my understanding correct? If yes, then the significance of the result is somewhat limited and seems heavily dependent on that assumption.

Regarding societal impacts, the paper does not flag any negative consequences of this research, which seems reasonable given the theoretical nature of the work.

**Paper Formatting Concerns:**

I don't have any concerns regarding formatting of the paper.

**Quality:**

3

**Strengths And Weaknesses:**

Quality:

S1 - The paper appears to be technically solid, providing rigorous theoretical results alongside empirical validation. The algorithms are described with clear theoretical guarantees: for complete graphs, the algorithm achieves an expected $(\min{2.06\beta, 3}+\epsilon)$-approximation using $\tilde O(n)$ space, and for general graphs an $O(\beta \log |E^-|)$-approximation with similar space usage. These results improve upon the best-known streaming trade-offs (e.g., beating the previous $3+\epsilon$ approximation barrier in dynamic streams). The proofs build on established methods (Pivot algorithm and LP-based rounding) and adapt them to incorporate prediction errors, which is intellectually sound. The experimental section is a notable strength: by demonstrating performance gains on real and synthetic data, the authors bolster confidence that the theoretical improvements translate into practice. The paper’s claims are well-supported by the analysis.

W1 - The reliance on an oracle predictor is a major weakness. The theory assumes a “$\beta$-level” predictor which may be non-trivial to obtain in practice. While this assumption is clearly stated and the algorithm is robust to predictor error, it does mean the quality of results hinges on an external learning component. The paper could further discuss how one might train or derive such a predictor in real-world scenarios.


Clarity:

S2 - The paper is well-written and organized, making a complex theoretical contribution accessible. The introduction clearly defines the correlation clustering problem and the streaming models, with helpful context on prior results and their limitations. Key concepts like the $\beta$-level predictor are explained intuitively (the predictor’s quality $\beta$ indicates how close the predicted clustering is to optimal), and the paper provides pointers to formal definitions in the text. The algorithms and theoretical results are presented with appropriate theorem statements and proofs.

W2 - One area for improvement in clarity could be a more explicit discussion on practical usage of the predictor (perhaps a short example of how a predictor might be learned or an intuitive illustration of its inputs/outputs)

Significance:


S3 - The contributions of this paper are significant, especially within the streaming and approximation algorithms community. Correlation clustering is a fundamental ML problem with many applications (e.g. community detection and image segmentation), and the streaming setting is increasingly relevant for large-scale data. Prior to this work, achieving better than a 3-approximation in a single-pass dynamic stream was an open challenge – this paper provides the first result to surpass that barrier by leveraging predictions. Moreover, the algorithm for general graphs significantly reduces memory requirements: it removes the dependence on storing all negative edges ($|E^-|$) by using predictions, which can be extremely beneficial when the graph is dense. In practical terms, if a reasonable predictor is available, one can now cluster streaming data with higher accuracy or using less memory than before. The paper also connects two areas (streaming algorithms and learning augmentation), demonstrating how machine learning predictions can be used to overcome traditional worst-case barriers.

W3 - One caveat on significance is that the improvements are conditional on predictor quality. If no reliable predictor exists, the method falls back to the known 3-approximation and offers no worst-case gain. Thus, the result is very meaningful assuming the predictor can be trained or provided. Additionally, while the paper simplifies implementation compared to a state-of-the-art 1.847-approx algorithm, the achieved approximation (≈2.06) is still higher than 1.847. There remains a gap to close if one hopes to reach near-optimal clustering in streaming.

Originality:


S4 - The idea of learning-augmented algorithms is a relatively recent trend, and this paper applies it in a novel way to the correlation clustering problem. To the best of my knowledge, there were no prior streaming algorithms for correlation clustering that incorporate predictions, so this represents a fresh contribution. The authors build on known techniques (like the Pivot algorithm by Ailon et al. and its streaming variants) but innovate by integrating a prediction oracle into the clustering process.

W4 - One could argue that the paper’s techniques are incremental in the sense that they extend existing algorithms with an oracle rather than inventing a completely new clustering method. The core idea is largely based on existing approaches, and the main theoretical results are derived from previous work, which limits the paper’s novelty. However, I acknowledge that making this extension successful and theoretically analyzable is non-trivial.

---

> ### Author Rebuttal · Authors · 2025-07-30
>
> We sincerely thank the reviewer for the helpful and constructive comments. Please find our detailed responses below.
> >[W1] The theory assumes a “$\beta$-level” predictor which may be non-trivial to obtain in practice. While this assumption is clearly stated and the algorithm is robust to predictor error, it does mean the quality of results hinges on an external learning component. The paper could further discuss how one might train or derive such a predictor in real-world scenarios.
>
> >[Q1] How do the authors envision obtaining such a predictor in practice? For instance, could it be trained on smaller samples or past instances of similar clustering problems, and how sensitive is the algorithm’s performance to mis-specification of $\beta$?
>
> >[W2] One area for improvement in clarity could be a more explicit discussion on practical usage of the predictor.
>
> We highlight that our algorithms do NOT need to take $\beta$ as input. The prediction quality $\beta$ is only used for **theoretical** analysis (e.g., approximation guarantees) of the algorithms.
>
> In practice, as stated in Lines 146-154, we may use ML models such as graph neural networks (GNNs) to learn pairwise distances from related networks defined on the same vertex set. These models can be trained to extract meaningful distances, e.g., by learning node embeddings that map vertices to a Euclidean space. Specifically, the model takes as input a related graph (e.g., a previous instance), represented via its adjacency matrix or feature matrix, and outputs an embedding vector for each vertex. The distances between these embeddings can then be computed, for example via inner products, and these naturally satisfy the triangle inequality, thereby serving as valid pairwise distances.
> >[Q2] The paper mentions the $1.847$-approximation insertion-only streaming algorithm by Cohen-Addad et al. (STOC 2024), noting that the new approach is much simpler and works in the dynamic setting. Could the authors elaborate on the performance gap here? For example, if we restrict to insertion-only streams, does the learning-augmented algorithm attain a similar 2.06 approximation, or could it be improved further? A brief discussion on whether the learning-augmented approach might close the gap to $1.847$ in insertion-only scenarios would be insightful.
>
> Thank you for the insightful question. Indeed, we have also developed an algorithm for complete graphs in insertion-only streams (see Appendix F), which differs from our algorithm in dynamic streams but achieves the same approximation ratio. We are also interested in whether our prediction model could help close the gap to $1.847$ in insertion-only streams. We note that the $1.847$-approximation algorithm by Cohen-Addad et al. (STOC 2024) is a purely combinatorial algorithm based on local search. Specifically, their algorithm needs to implement a local search routine during the stream via sampling, which introduces the main source of impracticality. Since their framework is fundamentally different from ours -- and in particular, their local search implementation does not appear to use distance information -- it is not clear how our distance-based predictions will help the algorithm. However, it is worth exploring whether new kinds of prediction models could be designed to make their algorithm more efficient. We leave this as future work.
> >[Q3] The general graph result uses an adapted $\beta$-level predictor. Is this essentially the same notion of predictor quality as in complete graphs, or does it require a different adaptation? It would be helpful if the paper clarified what adjustments are needed for the predictor in general graphs and whether obtaining such an adapted predictor is harder or easier than in the complete graph case.
>
> For general graphs, the adapted notion of a $\beta$-level predictor quality requires a slightly different formulation of prediction quality. Specifically, as described in Lines 142-145, the second condition becomes $\sum_{(u,v)\in E_H^+}w^\prime_{uv}d_{uv} + \sum_{(u,v)\in E^-}(1-d_{uv}) \le \beta \cdot \mathrm{OPT}$, where $H^+:=(V,E_H^+,w')$ is an $\varepsilon$-spectral sparsifier of $G^+=(V,E^+)$, which approximates all the cuts in $G^+$ within a $(1\pm \varepsilon)$ factor. This adaptation is introduced purely for theoretical analysis, as our algorithm for general graphs performs ball-growing on a sparsified graph. Thus, we can use the same method to obtain such a predictor as in the complete graph case; the only difference lies in the quality measurement.
>
> Furthermore, we note that our algorithm for general graphs (which uses the adapted predictor) can be extended to work under the same predictor definition as in the complete graph setting, for a broad class of graphs (i.e., bounded-degree graphs). We provide further details in Appendix G.4.
> >[Q4] In practice, do the authors observe any memory overhead due to the learning-augmented approach? If the space usage was tightened or if less space is available, how would the algorithm degrade?
>
> As is standard in the literature of learning-augmented algorithms, we do NOT account for the memory required to implement or store the predictor; our focus is on the algorithmic complexity **given** a predictor. That said, to address your concern, we reran some experiments and measured memory usage of training the binary classifier used in Table 3 on the SBM ($n = 1200$) dataset. Total memory usage for training and storing predictor information is ~341.3 MB.
>
> We also measured the memory usage of our algorithm given the predictor information and compared it to its non-learning counterpart:
> - CKLPU24: 8.1 MB
> - Algorithm 1 (given the predictor): 19.3 MB
>
> The results show that our algorithm uses slightly more memory. This is reasonable, as our algorithm computes two clusterings and thus needs to maintain two separate sets of data structures in order to output the clustering with lower cost. This overhead comes with the benefit of a better approximation guarantee. Finally, we remark that although we are not able to limit/tighten the space usage during execution, it remains comparable to that of the non-learning baseline.
> >[Q5] The algorithm is essentially formed by the truncation during the stream and the post-processing part. For the streaming phase, the idea is mainly from CKLPU [SODA’24] (cf. CM [NeurIPS’23]), and I do not see anything that is substantially novel. Am I understanding it correctly, or am I missing something? Also, the approximation factor is somewhat weak; to outperform the pivot-based approximation, there has to be a value of $\beta$ where you need predictions with very high quality, correct?
>
> Regarding your first question, your understanding is correct in some sense, as our algorithmic framework for complete graphs indeed builds upon CKLPU [SODA’24] and CM [NeurIPS’23]. However, we would like to emphasize that the analysis of our algorithms with predictions is non-trivial. We have provided a technical overview in Appendix C. Here, we highlight our main technical novelty: Our key observation is that the truncated version of the LP rounding algorithm by CMSY [STOC’15] is equivalent to a two-step procedure: first sampling a subgraph $G'$ based on the predictions, and then running a truncated version of the Pivot algorithm on $G'$. Our main technical contribution is to prove that 1) the cost of the pivot clusters produced by the truncated version of the LP rounding algorithm is at most $2.06\beta$ times the cost of the optimal solution; 2) the cost of the optimal solution on $G'$ does not differ from that on the original graph $G$ by a lot. In this way, our algorithms keep the space small while achieving an approximation ratio better than $3$ under good prediction quality.
>
> Regarding your second point, your understanding is correct **from the theory aspect**. However, as shown in our experimental results (see Section 6 and Appendix H), our algorithm performs much better in practice than the theoretical guarantee suggests. Moreover, our algorithms remain applicable in practice even when the predicted pairwise distances do not strictly satisfy the formal definition (Definition 3.1). As long as the distances are meaningful, they can be directly incorporated into our framework.
> >[L3] The ML oracle used by the algorithm seems somewhat artificial, and its practical implementation may be challenging. On a very high-level intuition, the way I see it is "Suppose we assume that we can predict similarities that could be used to obtain a good approximation of the optimal solution, then we have an approximation algorithm of that quality." Is my understanding correct?
>
> Yes, your understanding is correct. However, a commonly adopted assumption in prior work on learning-augmented algorithms is that the predictions are derived by applying slight perturbations to the optimal solution, which is already a good approximation to the optimal one.
>
> We believe that predicting pairwise distances in a graph is natural, as there are several practical situations where such predictions help exploit the clustering structure of related networks defined over the same vertex set (see examples in Lines 63-76). We also note that similar oracles for pairwise distances have been considered in prior work across different contexts [1, 2].
>
> [1] Yuko Kuroki, Atsushi Miyauchi, Francesco Bonchi, and Wei Chen. Query-Efficient Correlation Clustering with Noisy Oracle. NeurIPS 2024.
>
> [2] Sandeep Silwal, Sara Ahmadian, Andrew Nystrom, Andrew McCallum, Deepak Ramachandran, and Seyed Mehran Kazemi. KwikBucks: Correlation Clustering with Cheap-Weak and Expensive-Strong Signals. ICLR 2023.

---

> > ### Comment · Reviewer_S28r · 2025-07-31
> > **Answer to Author Rebuttal**
> >
> > Thank you for the detailed and thoughtful rebuttal. Your clarifications address virtually all of the points I raised.
> >
> > ---
> >
> > ### Predictor Realization and Practicality (W1/Q1/W2, L3)
> >
> > Your explanation that **β is not required by the algorithm** and that pairwise distances can be obtained from GNN-based embeddings (or similar) makes the paper clearer. I appreciate the examples of how an embedding model would supply triangle-inequality respecting distances. One follow-up question that would further help practitioners:
> >
> > > **Follow-up:** In your experiments, did you measure how clustering quality degrades as the learned distances become noisier (e.g., by systematically perturbing the embeddings or labels)? A small ablation might quantify the algorithm’s empirical robustness.
> >
> > ---
> >
> > ### Insertion-Only Streams (Q2)
> >
> > Thanks for pointing me to Appendix F; it is good to know an insertion-only version achieves the same ≈2.06 guarantee. I understand that blending your distance-based predictions with Cohen-Addad et al.’s local-search framework is non-trivial and is left as future work. That sounds reasonable.
> >
> > ---
> >
> > ### Adapted Predictor for General Graphs (Q3)
> >
> > Your explanation that the adapted definition is purely for the sparsifier analysis, and that the same predictor definition suffices on bounded-degree graphs, resolves my confusion; no further questions here.
> >
> > ---
> >
> > ### Memory Overhead (Q4)
> >
> > The concrete memory numbers (19.3 MB vs 8.1 MB) are helpful; they confirm the overhead is modest and aligned with the benefit in approximation. No further concerns on this point.
> >
> > ---
> >
> > ### Novelty and Technical Contribution (Q5)
> >
> > I appreciate the extra detail highlighting the two-step view of your algorithm (sampling subgraph H via predictions, then truncated Pivot on H) and the new charging argument that yields the 2.06β bound. This clarifies where the technical novelty lies beyond CKLPU \[’24].
> >
> > ---
> >
> > ## Impact on My Evaluation
> >
> > The rebuttal clarifies implementation details and strengthens the practical narrative. My overall assessment **remains positive and inclined towards acceptance** and I do not see a need to adjust the numerical ratings. If the follow-up question shows strong empirical robustness to noise, that would further reinforce the significance, but I will definitely not urge to show the experimental results. I recognize that the main contributions of this paper are theoretical, and I think it is significant.
> >
> > If there is something else that you want to clarify, or you think I might not have understood correctly, feel free to point it out.
> >
> > Thank you again for the constructive responses.

---

> > > ### Author Response · Authors · 2025-08-01
> > >
> > > We sincerely appreciate your prompt response and are glad to see that most of your concerns have been addressed. Please find our response to your follow-up question below.
> > > >**Follow-up:** In your experiments, did you measure how clustering quality degrades as the learned distances become noisier (e.g., by systematically perturbing the embeddings or labels)? A small ablation might quantify the algorithm’s empirical robustness.
> > >
> > > In our experiments, we have evaluated the effect of prediction quality on clustering quality, for both the noisy predictor and the spectral embedding predictor. Specifically, for the noisy predictor (see Figures 1(a-c) and Figures 2(a-c)), we adjust the prediction quality by adding more perturbations to the optimal solutions (recall that we have access to the optimal solutions in these cases). For the spectral embedding predictor (see Figure 2(d)), we adjust the prediction quality by changing the embedding dimension -- note that a larger embedding dimension indicates higher prediction quality (i.e., a smaller $\beta$). The empirical results show that our algorithm outperforms its non-learning counterpart across all datasets under good prediction quality, and does not perform worse when the predictions become noisier.
> > >
> > > Furthermore, we have conducted additional experiments to evaluate the empirical robustness of our algorithm with more natural binary classifier predictions. In particular, we randomly flip a portion of the predicted labels provided by the binary classifier and observe how the clustering quality degrades. The clustering costs ($\times$1e3) of our algorithm compared to its non-learning counterpart CKLPU24 on the SBM dataset ($n=1200$) are as follows:
> > > - CKLPU24: 105.3
> > > - Algorithm 1 (with no label flip): 35.9
> > > - Algorithm 1 (with 1% label flip): 56.9
> > > - Algorithm 1 (with 2% label flip): 67.1
> > > - Algorithm 1 (with 3% label flip): 88.8
> > > - Algorithm 1 (with 4% label flip): 99.7
> > > - Algorithm 1 (with 5% label flip): 105.3 -> the output of our algorithm becomes consistent with CKLPU24.
> > >
> > > The results align well with our theoretical guarantee and demonstrate the empirical robustness of our algorithm -- namely, that even when the learned distances become considerably noisy, the clustering quality does not degrade beyond that of the non-learning baseline. We hope this will further assist practitioners.

---

> > > > ### Comment · Reviewer_S28r · 2025-08-08
> > > > **Thank you for the further experimental results and answering the followup question**
> > > >
> > > > All my questions have been answered. After reading the comments from other reviewers and the comprehensive responses from the authors, I think there is no need to update my score. I reaffirm that I am inclined to accepting this paper.

---

### Official Review · Reviewer_WNQG · 2025-07-02

**Clarity:** 4
**Significance:** 3
**Originality:** 3
**Rating:** 5
**Confidence:** 3

**Summary:**

This paper studied learning-augmented correlation clustering in data streams for complete and general graphs. The goal is to cluster vertices of a graph arriving in a stream to minimize disagreements—positive edges that cross clusters and negative edges that fall within them.

The most commonly-studied version of the problem uses complete graphs, meaning that there are ${n \choose 2}$ edges in the graph, and each edge is annotated with either $(-)$ or $(+)$ labels. A more complicated version is on general graphs, in which we will lose the ability to ‘infer’ $(-)$ edges from the set of $(+)$ edges. In the offline setting, there are separations between the two cases: we could achieve 1.437-approximation in polynomial time for complete graphs, but we cannot obtain any $O(1)$-approximation for the version with general graphs under plausible complexity assumptions.

This paper studied semi-streaming algorithms for the paper in the presence of a learning-augmented oracle. The oracle in the paper predicts distances between vertex pairs. Ideally, if the predictions are always correct, the optimal solution will pay costs for negative edges with $(1-d_{u,v})$ and positive edges with $d_{u,v}$. The algorithm uses a parameter $\beta$ to represent a multiplicative approximation of $\text{OPT}$ for the cost induced by the above solution.

The main results of the paper are: $i).$ for the complete graph setting, an algorithm that achieves $(min{2.06β,3}+0.1)$-approximation in $O(n \text{polylog}(n))$ space; and $ii).$ for the general graph setting, an algorithm that achieves $O(\beta\log{|E^-|})$-approximation in $O(n \text{polylog}(n))$ space. All of these algorithms run in polynomial time for post-processing.

The paper further conducted some experiments, showing that the algorithm for complete graphs has competitive experimental performances. In particular, for synthetic dataset, the proposed algorithm has a competitive performance with algorithms based on agreement decompositions (which is known to be very good for experiments); for real-world datasets, the proposed algorithm outperforms other benchmarks.

**Questions:**

Some questions were asked in the weakness part of the paper. One additional comment: please make sure that the discussion in Table 1 is for the polynomial-time case only. If one only cares about the space complexity, as in the ‘theory aspect’ of streaming algorithms, you can simply store a cut sparsifier, enumerate all possible clusters (this takes $n^n$ time), and output the one with the smallest cost. This gives us a $(1+\varepsilon)$-approximation in semi-streaming space, although it is a very impractical algorithm. One of the main contributions of Assadi,  Khanna, and Putterman [STOC’25] is to show the approximation can be done in *polynomial time*.

**Ethical Concerns:**

["NO or VERY MINOR ethics concerns only"]

**Final Justification:**

As per the discussion with the authors and the AC, although I still have reservations about whether the prediction is "natural" enough, I believe the paper contains enough contributions for acceptance. I'm supportive of its acceptance.

**Limitations:**

This work mostly concerns the theoretical foundation of ML, and I do not see any immediate negative societal impacts.

**Quality:**

3

**Strengths And Weaknesses:**

My general opinion for this paper is supportive. The paper studied an important problem with learning-augmented oracles. It is nice to see that the learning-augmented oracle could work for both complete and general graphs. Learning-augmented algorithms are growing popular in the recent two years (e.g., by BDSW [APPROX’24]; CDGLP [NeurIPS’24]; MRTV [ITCS'25]), and although I have some reservations about whether the oracle prediction used in this paper is ‘natural’, it is good to see such addition to the literature.

The paper is also well-written, and despite the rapid developments of the literature in correlation clustering recently, the paper still did a decent job to emphasize the significance of their results.

On the flip side, I have the following main criticisms:
- Whether the oracle prediction used in this paper is ‘natural’: this is somehow subjective, and it is definitely not the most natural oracle one would come up with. The motivation of the oracle seems highly related to the metric LP used by CMSY [STOC’15]. In other words, somehow the model is designed to be tailored to the techniques the authors had in mind. This does not look natural to me.
- Most of the techniques used in this paper follow from existing work. For instance, in the algorithm for complete graphs, despite a rather involved description and analysis of the TRUNCATEDPIVOT algorithm, the analysis really follows from CKLPU [SODA’24]. The PAIRWISEDISS algorithm, together with the ball-growing algorithm for general graphs, does need some white-box adaptation of existing algorithms. However, these adaptations also do not seem to have much technical novelty.

Overall, I think despite the criticism, the merits of the paper still outweigh the flaws. I think much of my opinion was influenced by the fact that the paper contains algorithms for both complete and general-case correlation clustering and the landscape of the literature (which considers learning-augmented graph algorithms more practical now).

---

> ### Author Rebuttal · Authors · 2025-07-30
>
> We sincerely thank the reviewer for the helpful and constructive comments. Please find our detailed responses below.
> >[W1] Whether the oracle prediction used in this paper is ‘natural’: this is somehow subjective, and it is definitely not the most natural oracle one would come up with. The motivation of the oracle seems highly related to the metric LP used by CMSY [STOC’15]. In other words, somehow the model is designed to be tailored to the techniques the authors had in mind. This does not look natural to me.
>
> On the one hand, we believe that predicting pairwise distances in a graph is natural, as there are several practical situations where such predictions help exploit the clustering structure of related networks defined over the same vertex set (see examples in Lines 63-76). We also note that similar oracles for pairwise distances have been considered in prior work across different contexts [1, 2].
>
> On the other hand, as stated in Lines 146-154, while the second condition in our predictor definition (Definition 3.1) is a technical requirement for theoretical analysis, our algorithms remain applicable in practice even when the predicted pairwise distances do not strictly satisfy this condition. As long as the distances are meaningful, they can be directly incorporated into our framework. In fact, in certain scenarios, the predictions are derived by applying slight perturbations to the optimal solution, which is an assumption commonly adopted in prior work on learning-augmented algorithms. In such cases, we can define $d_{uv}=0$ if $u$ and $v$ are predicted to be in the same cluster, and $d_{uv}=1$ if they are predicted to be in different clusters. We conjecture that for many graphs, there exists a small $\beta$ such that the second condition in Definition 3.1 is satisfied. As shown in our experimental results (see Section 6 and Appendix H), our algorithm performs much better in practice than the theoretical guarantee suggests.
>
> [1] Yuko Kuroki, Atsushi Miyauchi, Francesco Bonchi, and Wei Chen. Query-Efficient Correlation Clustering with Noisy Oracle. NeurIPS 2024.
>
> [2] Sandeep Silwal, Sara Ahmadian, Andrew Nystrom, Andrew McCallum, Deepak Ramachandran, and Seyed Mehran Kazemi. KwikBucks: Correlation Clustering with Cheap-Weak and Expensive-Strong Signals. ICLR 2023.
> >[W2] Most of the techniques used in this paper follow from existing work. For instance, in the algorithm for complete graphs, despite a rather involved description and analysis of the TRUNCATEDPIVOT algorithm, the analysis really follows from CKLPU [SODA’24]. The PAIRWISEDISS algorithm, together with the ball-growing algorithm for general graphs, does need some white-box adaptation of existing algorithms. However, these adaptations also do not seem to have much technical novelty.
>
> We have provided a technical overview in Appendix C. Here, we highlight our main technical novelty as follows:
> 1. For complete graphs, the analysis of our algorithms with predictions is non-trivial. Our key observation is that the truncated version of the LP rounding algorithm by CMSY [STOC’15] is equivalent to a two-step procedure: first sampling a subgraph $G'$ based on the predictions, and then running a truncated version of the Pivot algorithms on $G'$. Our main technical contribution is to prove that 1) the cost of the pivot clusters produced by the truncated version of the LP rounding algorithm is at most $2.06\beta$ times the cost of the optimal solution; 2) the cost of the optimal solution on $G'$ does not differ from that on the original graph $G$ by a lot. In this way, our algorithms keep the space small while achieving an approximation ratio better than $3$ under good prediction quality.
> 2. For general graphs, we can incorporate our predictions into the existing ball-growing algorithm by performing ball-growing on a sparsifier $H^+$ for $G^+$, using the predictions as distance metrics. While this straightforward approach improves space complexity, it leads to an approximation ratio with an undesirable $\log n$ term, which is suboptimal when $n > |E^-|$. To address this issue, we refine the algorithm by introducing a branching strategy in the streaming phase, based on the tracked space usage of $|E^-|$. In this way, our algorithm keeps the space small while achieving a near $O(\log |E^-|)$-approximation under good prediction quality.
>
> >[Q] Please make sure that the discussion in Table 1 is for the polynomial-time case only. If one only cares about the space complexity, as in the ‘theory aspect’ of streaming algorithms, you can simply store a cut sparsifier, enumerate all possible clusters (this takes $n^n$ time), and output the one with the smallest cost. This gives us a $(1+\varepsilon)$-approximation in semi-streaming space, although it is a very impractical algorithm. One of the main contributions of Assadi, Khanna, and Putterman [STOC’25] is to show the approximation can be done in *polynomial time*.
>
> Thank you for the suggestion! Yes, we focus on polynomial-time semi-streaming algorithms. We will clarify this in the revised version.

---

> > ### Comment · Reviewer_WNQG · 2025-08-02
> >
> > Thanks for getting back to me about my questions. The technical novelty is clearer now, and I encourage the authors to add more discussions to separate the technical ideas used in this paper vs. the approaches the paper was built on.
> >
> > For the discussion about whether the oracle is natural: I read your response. While I do believe it makes sense, the justification is still quite subjective. In particular, I still think the oracle is too closely related to the technical requirements of the clustering LP, which makes it quite artificial. That being said, I agree that the practical performance is much better, and I'm not saying that this is a deal-breaker.
> >
> > Given that my initial assessment was positive, I'm keeping my score as it is. As I said, I'll be happy to see the paper gets accepted to the conference.

---

### Official Review · Reviewer_nHDZ · 2025-07-03

**Clarity:** 3
**Significance:** 3
**Originality:** 2
**Rating:** 4
**Confidence:** 3

**Summary:**

The paper studies the correlation cclustering problem in the streaming setting and proposes a new learning augmented algorithm for this problem. More specifically, given a $\beta$-level predictor, the algorithm can achieve an improved $(\min(2.06\beta, 3) + \epsilon)$-approximation in the complete graph case and an improved space in the general graph setting. Here the prediction information we want to get from the predictor is roughly about the pairwise distance on the graph (but more complicated, formally). Finally, the paper also presents an empirical evaluation on both synthetic and real-world datasetss, which demonstrates the advantage of the proposed algorithm.

**Questions:**

For the binary classifier in the experiment. What is the split of the training and test data (like, is the experiment setting similar to the setting where we use part of the dataset for training and run the evaluation on the remaining part)?

**Ethical Concerns:**

["NO or VERY MINOR ethics concerns only"]

**Quality:**

3

**Strengths And Weaknesses:**

Strengths:

- The technical contribution of this paper is solid. The paper gives a rigorous analysis (which is based on two previous theoretical papers) to show that, given a good predictor, there are better algorithms for the correlation clustering problem in both complete graph and general graph setting.

- The paper gives a somewhat detailed experiments section that suggests the advantage of the proposed algorithms.

- The organization of the paper is clear.

Weaknesses:

- I agree it is natural to consider predicting the pair-wise distance in a graph. But the formal definition of the $\beta$-level predictor is a bit artificial to me and it is not very clear to me whether this guarantee is easy to achieve in practice. Also, to get a better approximation ratio, the $\beta$ needs to be small (< 1.5).

- The algorithms proposed in this paper seem to be largely based on the previous purely theoretical work. From a technical perspective, the new algorithms are not super surprise to me (but I agree the theoretical work of this paper is solid).

---

> ### Author Rebuttal · Authors · 2025-07-30
>
> We sincerely thank the reviewer for the helpful and constructive comments. Please find our detailed responses below.
> >[Q] For the binary classifier in the experiment. What is the split of the training and test data (like, is the experiment setting similar to the setting where we use part of the dataset for training and run the evaluation on the remaining part)?
>
> For the binary classifier used in our experiments, we follow the standard ML practice of splitting the data into separate training (70%) and testing (30%) sets. We further remark that this predictor is used for datasets with available ground-truth communities. In this setting, the goal of Correlation Clustering aligns with that of community detection by treating edges between two vertices in the same (ground-truth) community as positive edges and edges between two vertices in different communities as negative edges. The predictions provided by the binary classifier on the entire dataset are then used as the pairwise distances in our Correlation Clustering algorithms.
> >[W1] I agree it is natural to consider predicting the pair-wise distance in a graph. But the formal definition of the $\beta$-level predictor is a bit artificial to me and it is not very clear to me whether this guarantee is easy to achieve in practice. Also, to get a better approximation ratio, the $\beta$ needs to be small (< 1.5).
>
> As stated in Lines 146-154, we may use ML models such as graph neural networks (GNNs) to learn pairwise distances from related networks defined on the same vertex set. These models can be trained to extract meaningful distances, for example, by learning node embeddings that map vertices to a Euclidean space. The distances between these embeddings naturally serve as pairwise distances and satisfy the triangle inequality (the first condition in our definition).
>
> While the second condition in our definition is a technical requirement for theoretical analysis, our algorithms remain applicable in practice even when the given pairwise distances do not strictly satisfy this condition. As long as the distances are meaningful, they can be directly incorporated into our framework. In fact, in certain scenarios, the predictions are derived by applying slight perturbations to the optimal solution, which is an assumption commonly adopted in prior work on learning-augmented algorithms. In such cases, we can define $d_{uv}=0$ if $u$ and $v$ are predicted to be in the same cluster, and $d_{uv}=1$ if they are predicted to be in different clusters. We conjecture that for many graphs, there exists a small $\beta$ such that the second condition in Definition 3.1 is satisfied. As our experimental results (see Section 6 and Appendix H) suggest, our algorithm performs much better in practice than the theoretical guarantee suggests.

---

> > ### Comment · Reviewer_nHDZ · 2025-08-06
> >
> > Thank you to the authors for the detailed response, which addresses most of my concerns. I will maintain my current opinion regarding the acceptance of the paper.
> >
> > Personally, I agree that predicting pairwise distances is a natural approach, and the proposed algorithm demonstrates good practical performance. However, I still think that the assumption of the oracle is too close to the technical requirements. It would be beneficial if there are some ways to relax this assumption from a theoretical perspective.

---

### Official Review · Reviewer_RLqw · 2025-07-04

**Clarity:** 3
**Significance:** 3
**Originality:** 3
**Rating:** 4
**Confidence:** 2

**Summary:**

The authors study streaming algorithms for Correlation Clustering, which use the predictions of pairwise distances between vertices provided by a predictor. Experimental results on synthetic and real-world datasets demonstrate the superiority. Experimental results on synthetic and real-world datasets demonstrate  the superiority of the proposed algorithms over their non-learning counterparts.

**Questions:**

1. The limitations are not discussed in the paper. This would be beneficial as it would present suggestions for future research that addresses the limitations of this work.

2. A more in-depth discussion of computational complexity, runtime, and performance under hardware limitations is needed to better assess the practicality of the method.

**Ethical Concerns:**

["NO or VERY MINOR ethics concerns only"]

**Limitations:**

1. I suggest that the author further enrich the experiments to verify the advantages of the model.

2. The organizational structure of the article needs to be further optimized, including some background knowledge and related work additions

**Paper Formatting Concerns:**

I haven't found any paper format problems yet.

**Quality:**

3

**Strengths And Weaknesses:**

Strengths:

1. The article has certain theoretical analysis and a clear explanation of the essence of the algorithm;

2. The proposed method is relatively novel.

3. The paper introduces an innovative idea that provides valuable insights into newly designed components, which could significantly benefit researchers in related fields.

Weaknesses:

1. I suggest that the author further enrich the experiments to verify the advantages of the model.

2. The organizational structure of the article needs to be further optimized, including some background knowledge and related work additions

3. The limitations are not discussed in the paper. This would be beneficial as it would present suggestions for future research that addresses the limitations of this work.

4. A more in-depth discussion of computational complexity, runtime, and performance under hardware limitations is needed to better assess the practicality of the method.

---

> ### Author Rebuttal · Authors · 2025-07-30
>
> We sincerely thank the reviewer for the helpful and constructive comments. Please find our detailed responses below.
> >[Q1] The limitations are not discussed in the paper. This would be beneficial as it would present suggestions for future research that addresses the limitations of this work.
>
> We conclude our limitations and potential future work as follows:
> 1. For complete graphs, our algorithm achieves a $(\min \\{2.06\beta, 3\\}+\varepsilon)$-approximation using $\tilde{O}(n)$ total space. There is a still gap between our approximation ratio and the best-known $(\alpha_\text{BEST}+\varepsilon)$-approximation [1] where $\alpha_\text{BEST}$ is the best approximation ratio of any polynomial-time classical/offline algorithm. It would be interesting to explore whether it is possible to achieve an $(\alpha_\text{BEST}+\varepsilon)$-approximation while using $\tilde{O}(n)$ **total** space (including the space for streaming and post-processing) under our prediction model.
> 2. For general graphs, our algorithm achieves an $O(\beta \log |E^-|)$-approximation using $\tilde{O}(n)$ total space. There is an extra $\beta$ factor in the approximation ratio compared to the best-known $O(\log |E^-|)$-approximation [2]. It would be interesting to explore whether it is possible to achieve a better-than-$O(\log |E^-|)$-approximation using the same space under our prediction model, thus further improving the best-known space-approximation tradeoff.
>
> We will include the above discussion in the revised version.
>
> [1] Sepehr Assadi, Sanjeev Khanna, and Aaron Putterman. Correlation Clustering and (De)Sparsification: Graph Sketches Can Match Classical Algorithms. STOC 2025.
>
> [2] Kook Jin Ahn, Graham Cormode, Sudipto Guha, Andrew McGregor, and Anthony Wirth. Correlation Clustering in Data Streams. ICML 2015.
> >[Q2] A more in-depth discussion of computational complexity, runtime, and performance under hardware limitations is needed to better assess the practicality of the method.
>
> We have conducted experiments to analyze the runtime of our algorithms (see Appendix H.4.1). The results show that our learning-augmented algorithms do not introduce significant time overheads. The slight increase in running time over its non-learning counterpart is due to the additional steps of querying the oracles and calculating the costs of two clusterings. These steps are both reasonable and acceptable.
>
> Since space efficiency is typically the primary concern in the streaming setting, we also measured the empirical memory usage of our algorithm given the predictor information and compared it to its non-learning counterpart:
> - CKLPU24: 8.1 MB
> - Algorithm 1 (given the predictor): 19.3 MB
>
> The results show that our algorithm uses slightly more memory. This is reasonable, as our algorithm computes two clusterings and thus needs to maintain two separate sets of data structures in order to output the clustering with lower cost. This overhead comes with the benefit of a better approximation guarantee.
> >[L1] I suggest that the author further enrich the experiments to verify the advantages of the model.
>
> Thank you for the valuable suggestion! We will consider further enriching the experiments to better evaluate the effectiveness and advantages of our proposed algorithms.
> >[L2] The organizational structure of the article needs to be further optimized, including some background knowledge and related work additions.
>
> Thank you for the valuable suggestion! Due to space limitations, we have placed further related work, additional technical preliminaries, and a technical overview including some background knowledge in Appendices A, B, and C. We will further optimize the organizational structure of the paper in the revised version.

---

> > ### Comment · Reviewer_RLqw · 2025-08-06
> >
> > The authors have effectively refuted my comments and added relevant experiments. I maintain my score.

---

### Decision · Program_Chairs · 2025-09-17

**Decision:**

Accept (poster)

**Comment:**

The paper studies learning augmented algorithms for the correlation clustering problem. By carefully combining an oracle which provides additional information, the authors are able to obtain a low space algorithm which have approximation quality comparable to the best algorithms without any space constraints. The reviewers did note that the prediction model maybe a bit unnatural, but were positive about the fact that this is an important first step. This was towards the top of my pile and I am recommending accept.